# Inverse Rendering using Multi-Bounce Path Tracing and Reservoir Sampling

**Yuxin Dai[1]\*, Qi Wang[2]\*, Jingsen Zhu[3]\*, Dianbing Xi[2], Yuchi Huo[2],
Chen Qian[4], Ying He[1]†**

[1]S-Lab, Nanyang Technological University [2]State Key Lab of CAD&CG, Zhejiang University
[3]Cornell University [4]SenseTime Research and Tetras.AI

## Abstract

We introduce MIRReS, a novel two-stage inverse rendering framework that jointly reconstructs and optimizes explicit geometry, materials, and lighting from multi-view images. Unlike previous methods that rely on implicit irradiance fields or oversimplified ray tracing, our method begins with an initial stage that extracts an explicit triangular mesh. In the second stage, we refine this representation using a physically-based inverse rendering model with multi-bounce path tracing and Monte Carlo integration. This enables our method to accurately estimate indirect illumination effects, including self-shadowing and internal reflections, leading to a more precise intrinsic decomposition of shape, material, and lighting. To address the noise issue in Monte Carlo integration, we incorporate reservoir sampling, improving convergence and enabling efficient gradient-based optimization with low sample counts. Through both qualitative and quantitative assessments across various scenarios, especially those with complex shadows, we demonstrate that our method achieves state-of-the-art decomposition performance. Furthermore, our optimized explicit geometry seamlessly integrates with modern graphics engines supporting downstream applications such as scene editing, relighting, and material editing. Project page: `https://brabbitdousha.github.io/MIRReS/`.

## 1 Introduction

Inverse rendering, the process of decomposing multi-view images into geometry, material and illumination, is a long-standing challenge in computer graphics and computer vision. The task is particularly ill-posed due to the inherent ambiguity in finding solutions to reproduce the observed image, especially when illumination conditions are unconstrained. Recent advancements in neural radiance fields (NeRFs) Mildenhall et al. (2021) and neural implicit surfaces (such as signed distance fields (SDFs)) Wang et al. (2021); Yariv et al. (2021) have inspired several works Boss et al. (2021a;b); Srinivasan et al. (2021); Zhang et al. (2021b; 2022; 2021a) that employ NeRF or SDF for scene representation. These methods often utilize auxiliary MLPs to predict materials or illumination. However, these MLP-based methods often suffer from limited network capacity and slow convergence, resulting in distorted geometries and inaccurate materials. In contrast, TensoIR Jin et al. (2023) employs a compact TensoRF-based representation Chen et al. (2022) with explicit second-bounce ray marching for more accurate indirect illumination. Despite their remarkable results, these implicit methods still have two **inherent disadvantages**: Firstly, they represent geometry as implicit density fields rather than triangular meshes, restricting their application in the graphics industry where triangular meshes are the most widely accepted digital assets. Secondly, while some methods do sample secondary rays for indirect illumination, they rely on radiance fields instead of physically-based rendering (PBR) to obtain second-bounce radiance, thus lacking the physical constraints necessary for precise material optimization.

---

\*Equal contribution.
†Corresponding author.

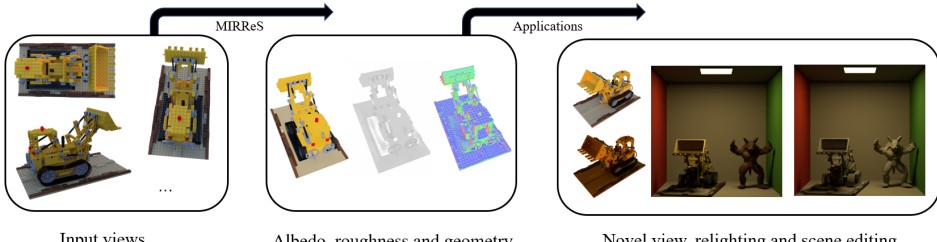

Figure 1: **Capabilities of MIRRes.** Given multi-view images of a 3D scene, our method jointly optimizes geometry, materials and lighting to achieve high-quality reconstructions. This facilitates applications including novel view synthesis, relighting, and scene editing.

To address these challenges, we propose *MIRReS*[1], a *mesh-based* two-stage inverse rendering framework that decomposes geometry, PBR materials, and illumination from multi-view images using physically-based *multi-bounce* path tracing. Our framework directly optimizes triangle meshes to enable applications such as scene editing, relighting, and material editing that are compatible with modern graphics engines and CAD software (see Fig. 1). The explicit use of triangular mesh representation also facilitates efficient path tracing with modern graphics hardware, making it possible to compute multi-bounce path tracing within acceptable timeframes. However, existing mesh-based neural inverse rendering methods, such as NVdiffrec-MC Hasselgren et al. (2022), often suffer from unstable geometry optimizations. These instabilities lead to artifacts such as holes and self-intersecting faces, which compromise the accuracy of ray-mesh intersections and make path tracing intractable, especially in multi-bounce cases where errors will accumulate recursively.

In particular, our paper consists of three key technologies: (a) **Mesh optimization.** Our approach incorporates a two-stage geometry optimization and refinement process (Section 3). (b) **Indirect illumination estimation.** We explicitly conduct physically-based multi-bounce path tracing with Monte Carlo integration, which enforces strict physical constraints on materials, thereby improving the accuracy of indirect illumination and relighting results. (c) **Convergence acceleration.** Recognizing that the Monte Carlo estimator necessitates a substantial sample count to maintain optimization accuracy, which inherently slows convergence, we leverage reservoir sampling Bitterli et al. (2020) for direct illumination, which reduces the required sample count while maintaining low noise levels. Meanwhile, coupled with a denoiser inspired by NVdiffrec-MC Hasselgren et al. (2022), our framework achieves a considerable convergence acceleration. To summarize, our contributions include:

1. We propose MIRReS, a physically-based inverse rendering framework that jointly optimizes the geometry, materials and lighting from multi-view input images, achieving state-of-the-art results in both decomposition and relighting.

2. Our method utilizes *multi-bounce path tracing* to provide a more accurate estimation of indirect illumination, successfully achieving promising decomposition results in the challenging highly-shadowed scenes.

3. Our method utilizes *Reservoir-based Spatio-Temporal Importance Resampling* for direct illumination, which can greatly reduce the required sample counts and accelerate the rendering process.

## 2 RELATED WORK

### 2.1 NEURAL SCENE REPRESENTATIONS

As an alternative to traditional representations (*e.g.* mesh, point clouds, volumes, *etc.* ), neural representations have achieved great success in novel view synthesis and 3D modeling. Neural radiance fields (NeRF) Mildenhall et al. (2021) uses MLPs to implicitly encode a

---

[1]Name taken from "Multi-bounce Inverse Rendering using Reservoir Sampling".

scene as a neural field of volumetric density and RGB radiance values, and uses volume rendering to produce promising novel view synthesis results. To address the limited expression ability and slow speed of the vanilla MLP representation, follow-up works leverage voxels Fridovich-Keil et al. (2022); Sun et al. (2022), hashgrids Müller et al. (2022), tensors Chen et al. (2022), polygon rasterization Chen et al. (2023), adaptive shells Wang et al. (2023b), *etc.* to achieve high-fidelity rendering result and real-time rendering speed. In addition to NeRF-based methods, 3D Gaussian Splatting (3DGS) Kerbl et al. (2023) proposes to use point-based 3D Gaussians to represent a scene, enabling fast rendering speed due to the utilization of rasterization pipeline, stimulating follow-up works on quality improvement or many other applications Yu et al. (2024); Tang et al. (2023a); Liang et al. (2023). Some other works also seek to combine neural and traditional representations, leveraging both strengths. For example, NeRF2Mesh Tang et al. (2023b) designs a two-stage reconstruction pipeline, which refines the textured mesh surface extracted from the NeRF density field to obtain delicate textured mesh recovery. In this work, we also employ a two-stage geometry optimization strategy combining neural implicit representation and triangle meshes.

## 2.2 Inverse rendering

The task of inverse rendering aims to estimate the underlying geometry, material and lighting from single or multi-view input images. Due to inherent ambiguity between the decomposed properties and the input images, inverse rendering is an extremely ill-posed problem. Some methods simply the problem under constrained assumptions, such as controllable lights Bi et al. (2020); Luan et al. (2021); Nam et al. (2018). Physically-based methods Li et al. (2018); Zhang et al. (2020); Jakob et al. (2022); Loubet et al. (2019) account for global illumination effects via differentiable light transports and Monte-Carlo path tracing. The emergence of neural representations has stimulated abundant neural inverse rendering frameworks Jin et al. (2023); Zhang et al. (2021b; 2023), which utilize neural fields as the positional functions of material and geometry properties, along with lighting as trainable parameters such as Spherical Harmonics (SH), Spherical Gaussian (SG), environment maps, *etc.* , and then jointly optimize them by rendering loss via differentiable rendering. However, neural fields also face challenges, such as low expressive capacity and high computational overhead caused by ray marching. Meanwhile, other methods also utilize explicit geometry representations, such as mesh Munkberg et al. (2022); Hasselgren et al. (2022) or 3D Gaussian Kerbl et al. (2023); Liang et al. (2023). Table 1 lists representative recent inverse rendering methods and compares their settings with our method. Our method is the first inverse rendering framework that supports multi-bounce raytracing to estimate indirect lighting more accurately.

Table 1: **Comparison between existing inverse rendering methods and our method.**

| Method | Geometry | Lighting | Indirect Lighting | Sampling |
|---|---|---|---|---|
| NeRFactor | Implicit | Environment | ✗ | N/A |
| TensoIR | Implicit | Ray tracing | ✓ | Importance sampling |
| NVdiffrec-MC | Mesh | Ray tracing | ✗ | Importance sampling |
| NeILF++ | Implicit | Implicit | ✓ | Stratified sampling |
| GS-IR | 3DGS | Split-sum | ✓ | N/A |
| Ours | Mesh | Multi-bounce Path tracing | ✓ | Reservoir sampling |

## 3 Method overview

This section outlines our proposed inverse rendering framework which utilizes multi-bounce raytracing and reservoir sampling. Given multi-view image captures of an object under unknown environment lighting conditions, along with their corresponding camera poses, our method jointly reconstructs the geometry, spatially-varying materials, and environment lighting. Unlike most recent approaches that use neural implicit geometry representations (*e.g.* NeRF or neural SDF), our method adopts a triangle mesh for an explicit geometry representation. This choice is crucial because optimizing mesh topology directly from multi-view images requires robust initialization. Inspired by NeRF2Mesh Tang et al. (2023b), we

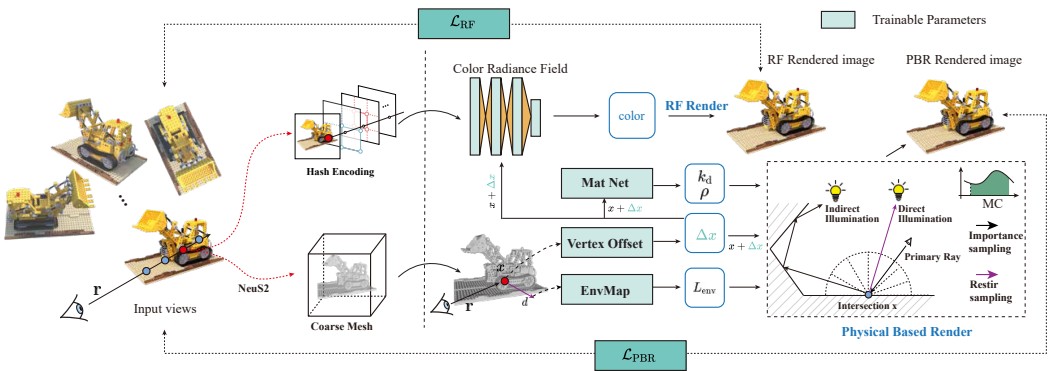

Figure 2: **Overview of our inverse rendering pipeline.** Our two-stage process starts with the extraction of a coarse mesh from a radiance field, followed by joint optimization of geometry, material, and lighting using physically-based rendering techniques. Key components such as multi-bounce path tracing, Monte Carlo integration, and reservoir sampling are highlighted to showcase their roles in enhancing the accuracy and efficiency of the reconstruction process.

employ a two-stage training process: initially, we train a neural radiance and signed distance field to extract a coarse mesh from the input images. The second stage, *which forms the core of our pipeline*, simultaneously optimizes scene material and lighting through our physically-based rendering and refines the mesh geometry. Fig. 2 provides an overview of our pipeline.

## 3.1 Stage 1: Radiance Field and Coarse Mesh Acquisition

The main purpose of this stage is to initialize a radiance field and geometry that will facilitate optimization in stage 2. We use an efficient off-the-shelf NeRF-based method (InstantNGP Müller et al. (2022)) to train a neural radiance field and a neural density field, with position $\mathbf{x}$ and view direction $\mathbf{d}$ as inputs, density $\sigma$ and radiance color $\mathbf{c}$ as outputs:

$$\sigma, \mathbf{f} = F_\sigma(\mathbf{x}), \quad \mathbf{c} = F_c(\mathbf{x}, \mathbf{d}, \mathbf{f}) \tag{1}$$

Although we can extract a coarse mesh from the density field $F_\sigma$ using the Marching Cubes algorithm, the NeRF-based volumetric representation inherently lacks geometric constraints. This limitation leads to the extraction of geometry with imperfections such as holes and sawtooth patterns, which adversely affect optimization in stage 2. Therefore, we additionally use NeuS2 Wang et al. (2023a)–a SOTA SDF-based reconstruction method–to extract the coarse mesh $\mathcal{M}_{\text{coarse}} = \{\mathcal{V}, \mathcal{F}\}$ (where $\mathcal{V}$ denotes vertices and $\mathcal{F}$ denotes faces). The density field $F_\sigma$ is then discarded, while $F_c$ continues to be optimized in stage 2 for geometry refinement.

## 3.2 Stage 2: Mesh Refinement and Intrinsic Decomposition

The goal of this stage is to refine the coarse mesh $\mathcal{M}_{\text{coarse}}$ obtained from stage 1 and to decompose material and environment lighting parameters into a fine mesh $\mathcal{M}_{\text{fine}}$.

**Rendering:** Starting with the extracted mesh $\mathcal{M}$ and a camera ray $\mathbf{r}(t) = \mathbf{o} + t\mathbf{d}$ from origin $\mathbf{o}$ in direction $\mathbf{d}$, we firstly use `nvdiffrast` Laine et al. (2020) to compute the ray-mesh intersection:

$$\mathbf{x} = \text{intersect}(\mathbf{r}, \mathcal{M}) \tag{2}$$

This differentiable process allows for gradient descent optimization and differentiable rendering. Our pipeline jointly employs two rendering methods: radiance field rendering and physically-based surface rendering, which will be used by the tasks of mesh refinement and intrinsic decomposition, respectively.

*Radiance field rendering:* With the appearance field $F_c$ from the NeRF network in the first stage, we produce the rendering result directly from the intersected surface point $\mathbf{x}$.

Unlike NeRF which uses ray marching and volume rendering, we directly feed the intersected surface point $\mathbf{x}$ from Eq. (2) into the appearance field to produce the ray color $C_{\text{RF}}(\mathbf{r})$:

$$C_{\text{RF}}(\mathbf{r}) = F_c(\mathbf{x}, \mathbf{d}, \mathbf{f}). \tag{3}$$

*Physically-based surface rendering:* Given the surface shading point $\mathbf{x}$, we render the shading color by the rendering equation Kajiya (1986), which is an integral over the upper hemisphere $\Omega$ at $\mathbf{x}$:

$$C_{\text{PBR}}(\mathbf{r}) = \int_{\Omega} L_i(\mathbf{x}, \omega_i) f_r(\mathbf{x}, \omega_i, \mathbf{d}, \mathbf{m})(\omega_i \cdot \mathbf{n}) d\omega_i, \tag{4}$$

where $L_i(\mathbf{x}, \omega_i)$ denotes the incident lighting from direction $\omega_i$, $\mathbf{m}$ denotes the spatially-varying material parameters at $\mathbf{x}$, $f_r$ denotes the bidirectional reflectance distribution function (BRDF), and $\mathbf{n}$ denotes the surface normal at $\mathbf{x}$.

In the context of multi-bounce path tracing, the incident light $L_i(\mathbf{x}, \omega_i)$ is the composition of direct light, which is the environment illumination in this work, and the indirect light:

$$\underset{\text{Direct environment illumination (Section 4.1)}}{\underbrace{\qquad\qquad}} \qquad \underset{\text{Indirect light (Section 4.2)}}{\underbrace{\qquad}}$$

$$L_i(\mathbf{x}, \omega_i) = \underbrace{V(\mathbf{x}, \omega_i)}_{\text{Direct Light Visibility}} L_{env}(\omega_i) + \boxed{L_{ind}(\mathbf{x}, \omega_i)}, \tag{5}$$

**Mesh refinement:** As mentioned in Section 1, implicit geometry representation may introduce bias and inaccuracy in indirect illumination estimation. Therefore, we opt to optimize a triangular mesh to represent scene geometry. DMTet Shen et al. (2021) used by NVdiffrec-MC Hasselgren et al. (2022) is an existing approach for direct mesh optimization, but it suffers from topological inconsistencies and geometric instability, which in turn affects the accuracy of path tracing. Instead, we opt for a stable and continuous optimization approach. Inspired by NeRF2Mesh Tang et al. (2023b), we assign a trainable offset $\Delta\mathbf{v}_i$ to each mesh vertex $\mathbf{v}_i \in \mathcal{V}$ to refine the geometry, and optimize them along with the appearance fields $F_c$ by minimizing the loss of radiance field rendering. Specifically, given a camera $s$ with known intrinsic and extrinsic parameters and its reference image $I_{\text{ref}}(s)$, we use radiance field rendering (Eq. (3)) to produce an image $I_{\text{RF}}(s)$, and then optimize $\Delta\mathbf{v}_i$ and the parameters of $F_c$ by minimizing the L2 loss between the rendering result and reference image:

$$\mathcal{L}_{\text{RF}} = \|I_{\text{RF}}(s) - I_{\text{ref}}(s)\|_2^2 \tag{6}$$

Similarly, we also use physically-based surface rendering (Eq. (4)) to produce an image $I_{\text{PBR}}(s)$. Owing to the differentiability of `nvdiffrast`'s ray-mesh intersection, the gradient of the L2 loss between the PBR rendering result and the reference image can be back-propagated to $\Delta\mathbf{v}_i$:

$$\mathcal{L}_{\text{PBR}} = \|I_{\text{PBR}}(s) - I_{\text{ref}}(s)\|_2^2 \tag{7}$$

In summary, $\Delta\mathbf{v}_i$ is jointly optimized by $L_{\text{RF}}$ and $L_{\text{PBR}}$ in stage 2, which refines the geometry from $\mathcal{M}_{\text{coarse}}$ to $\mathcal{M}_{\text{fine}}$. Since $\Delta\mathbf{v}_i$ is continually changing during the optimization and does not change the face topology of the mesh, our mesh refinement approach ensures geometry stability and ensures the feasibility of multi-bounce path tracing.

**Intrinsic decomposition:** Based on the mesh geometry, we now describe how we represent and optimize the spatially-varying material and environment lighting.

We adopt the physically-based BRDF model from Disney Burley & Studios (2012), which requires 2 material parameters: diffuse albedo and roughness. We encode the spatially-varying material parameters of the scene using a neural field $F_m$, which predicts the material parameters $\mathbf{m}$ given an input position $\mathbf{x}$: $\mathbf{m} = F_m(\mathbf{x})$. We implement $F_m$ as a small MLP with a multi-resolution hashgrid Müller et al. (2022). The predicted $\mathbf{m}$ is a 4-channel vector, which will be further split by channel into the diffuse albedo (3) and roughness (1).

Following NVdiffrec-MC Hasselgren et al. (2022), we represent the environment lighting as an HDR environment map with 256×512 pixels, where all pixel colors are trainable parameters.

## 4 Direct and indirect lighting

As described in Eq. (5), the incident lighting in the rendering equation is divided into two components: direct and indirect. These are estimated through reservoir sampling (Section 4.1) and multi-bounce raytracing (Section 4.2), respectively.

### 4.1 Direct lighting using reservoir sampling

Substituting the direct light part of Eq. (5) into Eq. (4) gives the rendering equation of direct light:

$$C_{\text{PBR}}^{dir}(\mathbf{r}) = \int_{\Omega} \underbrace{V(\mathbf{x}, \omega_i) \, L_{env}(\omega_i) \, f_r(\mathbf{x}, \omega_i, \mathbf{d}, \mathbf{m})(\omega_i \cdot \mathbf{n})}_{\equiv f(\omega_i)} \, d\omega_i \approx \frac{1}{N} \sum_{i=1}^{N} \frac{f(\omega_i)}{p_{dir}(\omega_i)}, \quad (8)$$

In the second part of the equation, we estimate the integral with Monte Carlo integration, with $N$ samples drawn from a particular distribution $p_{dir}(\omega_i)$, where only $V(\mathbf{x}, \omega_i)$ and $p_{dir}(\omega_i)$ are unknown, while the remaining terms are either analytically determined or trainable parameters. Therefore, determining $V(\mathbf{x}, \omega_i)$ and finding an appropriate $p_{dir}(\omega_i)$ is key to the estimation.

*Visibility estimation:* Unlike implicit-based methods like TensoIR Jin et al. (2023), which estimate the visibility function $V(\mathbf{x}, \omega_i)$ by the transmittance function in volume rendering, our method can directly estimate $V(\mathbf{x}, \omega_i)$ by a ray-mesh intersection. We implement our ray-mesh intersection algorithm with customized CUDA kernels, which significantly improves computational efficiency and enables us to increase the sample count for a more precise and low-variance estimation. Please refer to the appendix for implementation details.

*Reservoir sampling:* To reduce the variance of direct lighting estimation (*i.e.* to reduce rendering noise, see Fig. 3), we utilize reservoir sampling Bitterli et al. (2020), an advanced resampled importance sampling (RIS) technique Talbot (2005) to determine the appropriate $p_{dir}(\omega_i)$. According to the multiple importance sampling (MIS) theory Veach & Guibas (1995), the variance of the Monte Carlo estimator will reduce when $p_{dir}(\omega_i)$ is closer to the integrand $f(\omega_i)$. A distribution $p_{dir}(\omega_i) \propto L_{env}(\omega_i) f_r(\mathbf{x}, \omega_i, \mathbf{d}, \mathbf{m})$ will be an ideal solution, but it is impossible to analytically sample from such a probability distribution as it does not have a closed-form expression.

Reservoir   w/o Reservoir

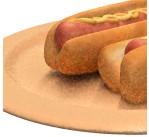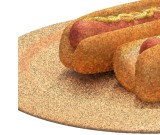

Figure 3: Comparison on rendering noise with or without reservoir sampling with sample count 1.

To address this, Resampled Importance Sampling (RIS) Talbot (2005) provides a more advanced technique to approximate the *target distribution* $p_{\text{dir}}(\omega_i) \propto L \cdot f$. Firstly, an easy-to-sample *proposal distribution* $q_{dir}(\omega_i)$ is chosen, from which $m$ samples $\mathcal{S} = \{\omega_1, ..., \omega_m\}$ are generated as candidates. In our implementation, we choose $q_{dir}(\omega_i) \propto L_{env}(\omega_i)$, which can be directly sampled from our environment map. Then, we evaluate $\hat{p}_{dir}(\omega_i) = L(\omega_i) \cdot f(\omega_i)$ for each candidate sample $\omega_i \in \mathcal{S}$, and assign a weight $\gamma_i = \frac{\hat{p}_{dir}(\omega_i)}{q_{dir}(\omega_i)}$ to it. Finally, we resample from $\mathcal{S}$ according to the weight $\gamma_i$. Weighted-averaging the sampled results after repeating for $N$ times forms an $N$-sample RIS estimator of

$$C_{\text{PBR}}^{dir}(\mathbf{r}) \approx \frac{1}{N} \sum_{i=1}^{N} \left( \frac{f(\omega_i)}{\hat{p}_{dir}(\omega_i)} \frac{1}{m} \sum_{s=1}^{m} \overbrace{\frac{\hat{p}_{dir}(\omega_s)}{q_{dir}(\omega_s)}}^{\text{RIS sample weight } (i.e. \ \gamma_s)} \right). \quad (9)$$

We note that the hat notation $\hat{p}_{dir}(\omega_i)$ means it is *not necessarily a normalized PDF*, since the normalization term can be reduced in Eq. (9). Intuitively, this sampler behaves as if weighting the RIS sample weight $\gamma_i$ to adjust for the difference between the proposal distribution $q_{dir}$ and the target distribution $p_{dir}$.

In addition, we also exploit spatial reuse and temporal reuse, incorporating samples from neighboring pixels and previous frames as candidates in Eq. (9). Please refer to our appendix for details. As in Fig. 3, our reservoir sampling strategy significantly reduces the rendering noise under the same sample count compared to the standard Monte Carlo estimator.

### 4.2 Indirect lighting using multi-bounce path tracing

Similar to Eq. (8), we can also give the rendering equation of indirect light and estimate it by Monte Carlo integration:

$$C_{\text{PBR}}^{ind}(\mathbf{r}) = \int_{\Omega} L_{ind}(\mathbf{x}, \omega_i) f_r(\mathbf{x}, \omega_i, \mathbf{d}, \mathbf{m})(\omega_i \cdot \mathbf{n}) d\omega_i, \tag{10}$$

$$\approx \frac{1}{N} \sum_{k=1}^{N} \frac{L_{ind}(\mathbf{x}, \omega_i^k) f_r(\mathbf{x}, \omega_i^k, \mathbf{d}, \mathbf{m}) \left( \omega_i^k \cdot \mathbf{n} \right)}{p_{ind} \left( \omega_i^k \right)}. \tag{11}$$

In Eq. (11), after sampling $N$ second-bounce rays $\mathbf{r}_k(t) = \mathbf{x} + t\omega_i^k$ from an appropriate probability distribution $p_{ind}(\omega_i)$, we need to estimate their radiance values $L_{ind}(\mathbf{x}, \omega_i)$ to complete the Monte Carlo estimation. The $p_{ind}(\omega_i)$ of indirect lighting is straightforward: we apply multiple importance sampling combining light importance sampling and GGX material importance sampling Heitz (2018), similar to NVdiffrec-MC Hasselgren et al. (2022).

Estimating $L_{ind}(\mathbf{x}, \omega_i)$ is relatively complicated. Most existing inverse rendering methods considering indirect lighting adopt an implicit strategy to estimate indirect illumination, using neural radiance fields to cache the outgoing radiance values of $\mathbf{r}_k$. The radiance values of $\mathbf{r}_k$ are estimated by the standard volume rendering process in NeRF. The disadvantage of this strategy is obvious: the accuracy of indirect lighting is determined by the neural radiance field without any physical constraints. NeRF models inevitably contain estimation errors (especially in scenes with high-frequency details), so that the indirect lighting estimation will be biased and inaccurate.

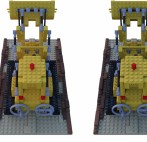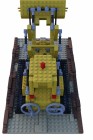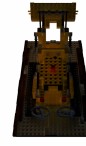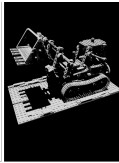

(a) Full    (b) Direct    (c) Indirect    (d) Light Visibility

Figure 4: Our rendering results of direct (b), indirect (c), and full (a) lighting in the *Lego* scene. Note that the sharp light visibility in (d) demonstrates the accuracy of our path-tracing rendering model and our reconstruction geometry.

In contrast, physically-based rendering methods conduct multi-bounce path tracing to produce an unbiased estimation of $L_{ind}(\mathbf{x}, \omega_i)$. However, due to the low performance of implicit representations, recursively performing multi-bounce path tracing leads to intractable computation. Therefore, the aforementioned implicit strategy can be regarded as a compromise on the computation costs.

Benefiting from our efficient mesh-based representation, we can directly perform path tracing to estimate $L_{ind}(\mathbf{x}, \omega_i)$ as shown in Fig. 4. We first sample a new ray $\mathbf{r}_{ind}(t) = \mathbf{x} + t\mathbf{d}_{ind}$ starting from $\mathbf{x}$ as the second bounce, and then we trace $\mathbf{r}_{ind}(t)$ and intersect it with the mesh at point $\hat{\mathbf{x}}$. The indirect lighting is estimated by:

$$L_{ind}(\mathbf{x}, \omega_i) = C_{\text{PBR}}(\mathbf{r}_{ind}) = \int_{\Omega} L_i(\hat{\mathbf{x}}, \omega_i) f_r(\hat{\mathbf{x}}, \omega_i, \mathbf{d}_{ind}, \hat{\mathbf{m}})(\omega_i \cdot \hat{\mathbf{n}}) d\omega_i. \tag{12}$$

Note that Eq. (12) is a recursive computation. In practice, we only consider the first three bounces, which balances the computation costs and accuracy. It is also worth mentioning that we detach the gradients of indirect rays due to the limited GPU memory.

## 5 Experiments

### 5.1 Overview

*Implementation and training details:* We implement MIRReS using Pytorch framework Paszke et al. (2019) with CUDA extensions in `SLANG.D` Bangaru et al. (2023). We

customize CUDA kernels in our rendering layer to perform efficient reservoir sampling and multi-bounce path tracing. We also utilize `nvdiffrast` Laine et al. (2020) for differentiable ray-mesh intersection. We run our training and inference on a single NVIDIA RTX 4090 GPU, with the entire two-stage training process taking approximately 4.5 hours.

Training follows the structure described in Section 3. The first stage' is identical to the training of InstantNGP Müller et al. (2022) and NeuS2 Wang et al. (2023a), for which we refer to their papers for details such as training losses. Our model is trained by rendering loss (Eqs. (6) and (7)) along with several regularization terms, which are fully specified in the appendix.

*Datasets:* We evaluate our method on two benchmark datasets for inverse rendering: (1) TensoIR synthetic dataset Jin et al. (2023), which provides ground-truth geometry, material parameters, and relighted images, and (2) Objects-with-Lighting (OWL) real dataset Ummenhofer et al. (2024), from which we select four scenes (*Antman*, *Tpiece*, *Gamepad*, and *Porcelain Mug*) containing ground-truth relighted images and environment maps. All objects exhibit spatially-varying materials and complex global illumination effects such as diffuse inter-reflections and specular highlights, making inverse rendering particularly challenging.

*Metrics:* To assess geometry reconstruction, we use the Mean Angular Error (MAE) for normal estimation and the Chamfer Distance (CD) for mesh accuracy. For intrinsic decomposition, we evaluate albedo reconstruction quality, novel view synthesis of physically-based surface rendering, and relighting performance. We use the widely-used Peak Signal-to-Noise Ratio (PSNR), Structural Similarity Index Measure (SSIM), and Learned Perceptual Image Patch Similarity (LPIPS) as evaluation metrics. It is worth mentioning that due to the inherent ambiguity between the scale of albedo and illumination, we apply a scaling strategy similar to TensoIR Jin et al. (2023). For the TensoIR dataset, each RGB channel of all albedo results is scaled by a global scalar; for the OWL dataset, the exposure level for each relighting result is also scaled.

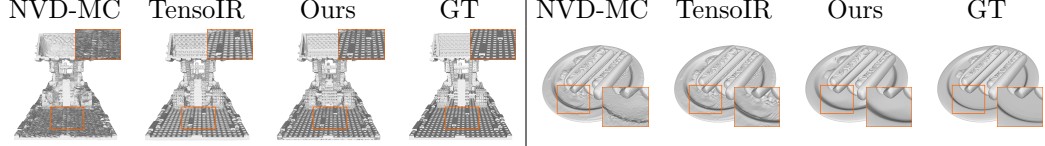

Figure 5: **Qualitative comparison of the reconstructed mesh on the TensoIR dataset.** Zoom in for details.

## 5.2 COMPARISONS

We compare our method against state-of-the-art inverse rendering techniques, including TensoIR Jin et al. (2023) (implicit-based), GS-IR Liang et al. (2023) (3D Gaussian Splatting-based), NVdiffrec-MC Hasselgren et al. (2022) (mesh-based, denoted as NVD-MC in figures and tables). We demonstrate that MIRRes outperforms these baselines in both qualitative and quantitative evaluations.

Table 2: **Quantitative comparison of reconstructed geometry, albedo and relighting, novel view synthesis on the TensoIR dataset.** "CD" denotes Chamfer distance, while "N-MAE" denotes normal MAE. Metrics are averaged over all testing images in all dataset scenes. We highlight the best , second-best , third-best results, accordingly.

| Method | Geometry | | Albedo | | | Relighting | | |
|---|---|---|---|---|---|---|---|---|
| | CD↓ | N-MAE↓ | PSNR↑ | SSIM↑ | LPIPS↓ | PSNR↑ | SSIM↑ | LPIPS↓ |
| NVD-MC | 0.073 | 5.050 | 28.875 | 0.957 | 0.082 | 27.810 | 0.907 | 0.110 |
| TensoIR | 0.083 | 4.100 | 29.275 | 0.950 | 0.085 | 28.580 | 0.944 | 0.081 |
| GS-IR | N/A | 4.948 | 30.286 | 0.941 | 0.084 | 24.374 | 0.885 | 0.096 |
| Ours | 0.056 | 3.305 | 32.348 | 0.970 | 0.054 | 32.363 | 0.965 | 0.055 |

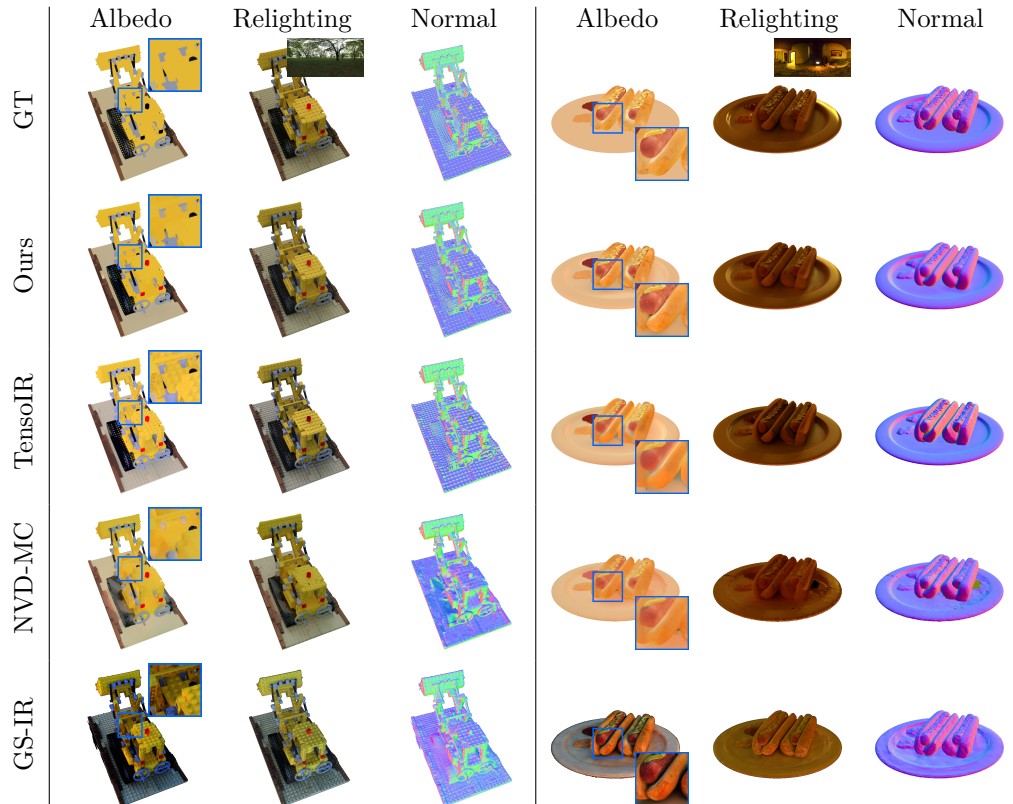

Figure 6: **Qualitative comparison of albedo, relighting and normal results on the TensoIR dataset.** The environment map used for relighting is shown in the inset. Images are rendered in high resolution to facilitate detailed examination when zoomed in.

**Comparison on geometry reconstruction:** The first two columns of Table 2 report the quantitative comparisons of the normal MAE and the chamfer distance between the reconstructed mesh and the ground truth mesh. On average, our method achieves 23.3% lower mesh error and a 19.4% lower normal error compared to the second-best baseline. We also provide qualitative results of reconstructed meshes in Fig. 5 and normal maps in the last row of Fig. 6. Note that GS-IR lacks mesh extraction, making Chamfer distance inapplicable. TensoIR struggles with high-frequency details and sharp edges due to implicit density field limitations. NVdiffrec-MC suffers from artifacts such as holes and uneven surfaces, especially in reflective objects (e.g., the dish in the *Hotdog* scene). MIRRes, leveraging a two-stage approach, produces refined meshes with higher quality.

**Comparison on decomposition and relighting results:** We perform a comprehensive comparison of the decomposed albedo, environment lighting and novel view synthesis results by PBR rendering. In this paragraph, we'll report both qualitative and quantitative results on TensoIR dataset and OWL dataset, respectively.

*TensoIR dataset:* Qualitatively, as shown in Fig. 6, our method produces superior decomposition results to all baselines. TensoIR and GS-IR have difficulties in correctly decomposing the lighting effects from the materials, leading to artifacts such as baked-in shadows on the albedo texture. NVdiffrec-MC produces suboptimal results due to their unstable geometry reconstruction, resulting in hole-like artifacts. On the other hand, thanks to our physically-based design to capture direct and indirect lighting, we successfully produce high-quality recovered material, eliminating the highly challenging shadow-like artifacts on the material textures. We also produce more realistic relighting results, including more accurate shadows, specular highlights and inter-reflections. In contrast, due to the baked-in shadows, baseline methods (like TensoIR) suffer from incorrect shadows and highlights in their relighting results. Please refer to the appendix for comprehensive per-scene decomposition results.

Further, we also report quantitative comparisons in Table 2, where our method achieves significant advantages over baselines in all results.

*Objects-with-Lighting (OWL) dataset:* We report quantitative comparisons on relighting and novel view synthesis in Table 3. Note that we use the NeuS Wang et al. (2021)-reconstructed mesh provided by the dataset as the initial coarse mesh in stage 1, rather than using NeuS2, which we empirically find better quality. We also incorporate metallic as an additional learnable channel in the output of the material network for this dataset. Our method achieves the best scores in all metrics, outperforming all the baselines by a large margin. We also provide per-scene qualitative comparisons in the appendix.

Table 3: **Quantitative comparison of novel view synthesis and relighting on the Object-with-Lighting dataset.** Metrics are averaged across all testing images from 4 selected scenes: *Antman, Tpiece, Gamepad, Porcelain Mug.*

| Method | Relighting | | | Novel View Synthesis | | |
|---|---|---|---|---|---|---|
| | PSNR↑ | SSIM↑ | LPIPS↓ | PSNR↑ | SSIM↑ | LPIPS↓ |
| NVD-MC | 21.110 | 0.970 | 0.066 | 34.409 | 0.967 | 0.059 |
| TensoIR | 26.382 | 0.966 | 0.038 | 37.127 | 0.985 | 0.045 |
| GS-IR | 18.761 | 0.101 | 0.314 | 30.527 | 0.793 | 0.096 |
| Ours | 28.827 | 0.977 | 0.031 | 38.223 | 0.986 | 0.030 |

## 5.3 ABLATION STUDIES

We conduct ablation studies to validate the effectiveness of our two key components: reservoir sampling and multi-bounce ray tracing. Specifically, we quantitatively compare the albedo PSNR in Table 4 using models with and without these components. The results show that both contribute to an increase in PSNR, with the full model achieving the best overall performance.

Table 4: **Ablation studies on reservoir sampling and multi-bounce raytracing.**

| Reservoir | Multi-bounce | Albedo PSNR | Relighting PSNR |
|---|---|---|---|
| ✗ | ✗ | 31.950 | 28.992 |
| ✓ | ✗ | 32.529 | 31.239 |
| ✗ | ✓ | 33.752 | 31.694 |
| ✓ | ✓ | 34.348 | 33.788 |

We also perform additional ablation studies on *Indirect illumination*, *Number of SPPs*, and *Neural radiance field rendering*, with detailed results provided in the appendix.

## 6 CONCLUSION

We introduce a two-stage, physically-based inverse rendering framework that jointly reconstructs and optimizes explicit geometry, materials, and illumination from multi-view images. In the first stage, we train a neural radiance field and extract a coarse mesh as the initial geometry. In the second stage, we refine this mesh geometry using trainable offsets while optimizing materials and illumination through a physically-based inverse rendering model that leverages multi-bounce path tracing and Monte Carlo integration. To improve convergence in Monte Carlo rendering, we integrate sampling with multi-importance sampling, reducing variance and maintaining low rendering noise even at low sample counts. Our experiments, particularly in scenes with complex shadows, demonstrate that our method achieves state-of-the-art performance in scene decomposition, effectively recovering shape, material, and lighting.

## ACKNOWLEDGMENTS

This work was supported in part by the Ministry of Education, Singapore, under its Academic Research Fund Grants (MOE-T2EP20220-0005 & RT19/22) and the RIE2020 Industry Alignment Fund–Industry Collaboration Projects (IAF-ICP) Funding Initiative, as well as cash and in-kind contribution from the industry partner(s).

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

## APPENDIX

## A    IMPLEMENTATION AND TRAINING DETAILS

**Visibility estimation.**    We here describe our visibility estimation based on an explicit ray-mesh intersection computation (as described in Section 4.1). Based on our mesh-based geometry, our method can directly determine $V(\mathbf{x}, \omega_i)$ by a ray-mesh intersection in our path tracing framework. With the intersection point $\mathbf{x}$ obtained from `nvdiffrast`, we sample an outgoing direction $\omega_i$ to construct the visibility test ray $\mathbf{r}_i(t) = \mathbf{x} + t\omega_i$. Then, we conduct a ray-mesh intersection test to determine whether the ray is occluded. $V(x, \omega_i)$ will be 0 if $\mathbf{r}_i(t)$ is occluded, otherwise 1. Benefiting from our mesh-based representation, we implement a linear BVH (LBVH Karras (2012)) using CUDA kernels to significantly accelerate the ray-mesh intersection calculation. Our LBVH is updated in each iteration to match the mesh refinement.

**Network details.**    In stage 1, the density and appearance fields $F_\sigma, F_c$ follow the standard Instant-NGP configuration Müller et al. (2022), using a hash grid with 16 levels, 2 feature dimensions per entry, a coarsest resolution of $N_{\min} = 16$, and a finest resolution of $N_{\max} = 2048$, followed by a 4-layer MLP with 64 hidden channels. In stage 2, the material network $F_m$ uses the same hash grid configuration, followed by a 2-layer MLP with 32 hidden channels.

**Training details.**    Apart from the rendering losses (Eqs. (6) and (7)) mentioned in the main text, we also add several additional regularizations to stabilize the training, described as follows:

To prevent drastic changes in vertex offset $\mathbf{\Delta v}$ during optimization, we apply the Laplacian smooth loss and vertices offset regularization loss from NeRF2Mesh Tang et al. (2023b):

$$\mathcal{L}_{\text{smooth}} = \sum_i \sum_{j \in X_i} \frac{1}{|X_i|} \left\| (\mathbf{v}_i + \mathbf{\Delta v}_i) - (\mathbf{v}_j + \mathbf{\Delta v}_j) \right\|^2, \tag{13}$$

$$\mathcal{L}_{\text{offset}} = \sum_i \left\| \mathbf{\Delta v}_i \right\|^2, \tag{14}$$

where $X_i$ is the set of adjacent vertex indices of $\mathbf{v}_i$.

We also apply the smoothness regularizers for albedo $\boldsymbol{k_d}$, roughness $\rho$, and normal $\mathbf{n}$ proposed by NVdiffrec-MC Hasselgren et al. (2022) for better intrinsic decomposition:

$$\mathcal{L}_{\boldsymbol{k}} = \frac{1}{|X|} \sum_{x_i \in X} |\boldsymbol{k}(\mathbf{x}_i) - \boldsymbol{k}(\mathbf{x}_i + \boldsymbol{\epsilon})|, \quad \boldsymbol{k} \in \{\boldsymbol{k_d}, \rho, \mathbf{n}\}, \tag{15}$$

where $X$ is the set of world space positions on the surface, and $\epsilon$ is a small random offset vector.

Additionally, for better disentangling material parameters and light, we adopt the same monochrome regularization term of NVdiffrec-MC Hasselgren et al. (2022):

$$\mathcal{L}_{\text{light}} = |\text{Y}(\mathbf{c}_d + \mathbf{c}_s) - \text{V}(I_{\text{ref}})|, \tag{16}$$

where $\mathbf{c}_d$ and $\mathbf{c}_s$ are the demodulated diffuse and specular lighting terms, $Y(\mathrm{x}) = (\mathrm{x}_r + \mathrm{x}_g + \mathrm{x}_b)/3$ is a luminance operator, $V(\mathbf{x}) = \max(\mathbf{x}_r, \mathbf{x}_g, \mathbf{x}_b)$ is the HSV value component. For more details and discussions of this loss, please refer to NVdiffrec-MC Hasselgren et al. (2022).

**Denoiser.** We adopt the Edge-Avoiding À-Trous Wavelet Transform (EAWT) Dammertz et al. (2010) as an efficient and stable denoiser. This method uses a wavelet decomposition, where the input $c_0(p)$ is iteratively smoothed using a $B_3$-spline kernel $h$ Murtagh (1998). At each level $i$, the signal is decomposed into a residual $c_{i+1}$ as follows:

$$c_{i+1}(p) = c_i(p) * h_i, \tag{17}$$

where $h_i$ expands its support by inserting $2^{i-1}$ zeros between coefficients at each step, ensuring computational efficiency.

To preserve edges during smoothing, a data-dependent weighting function $w(p, q)$ is introduced. This function incorporates information from the ray-traced input image (rt), normal buffer ($n$), and position buffer ($x$):

$$w(p, q) = w_{\text{rt}} \cdot w_n \cdot w_x, \tag{18}$$

where

$$w_{\text{rt}}(p, q) = \exp\left(-\frac{|I_p - I_q|^2}{\sigma_{\text{rt}}^2}\right) \tag{19}$$

is based on color differences between pixels $p$ and $q$. The $\sigma$-parameters control the sensitivity to variations. These weights ensure that the filter adapts to the scene structure, preventing excessive edge blurring.

The computation of $c_{i+1}(p)$ involves a normalization factor $k$, defined as:

$$k = \sum_{q \in \Omega} h_i(q) \cdot w(p, q). \tag{20}$$

Using $k$, $c_{i+1}(p)$ is then computed as:

$$c_{i+1}(p) = \frac{1}{k} \sum_{q \in \Omega} h_i(q) \cdot w(p, q) \cdot c_i(q). \tag{21}$$

We perform three iterations of this process, balancing accuracy and computational efficiency.

**Spatial-temporal Reuse.** Eq. (9) can be implemented in a straightforward way by generating and storing all $m$ candidate samples before selecting a final sample. However, this approach is computationally demanding. To address this, we utilize weighted reservoir sampling (WRS) Chao (1982) , which transforms RIS into a streaming way. That is, we maintain a reservoir structure $r = \{y^r, \gamma_{\text{sum}}^r, M^r\}$ for each pixel, where $y^r$ is the selected sample, $\gamma_{\text{sum}}^r$ is the sum of the weights, and $M^r$ is the number of samples seen so far. When a new candidate $(\omega_s, \gamma_s)$ comes in, we update $\gamma_{\text{sum}}^r, M^r$ and decide whether to select the new sample based on the ratio of $\frac{\gamma_s}{\gamma_{\text{sum}}^r}$.

After the final sample is selected for each primary ray (or pixel), we exploit both spatial reuse and temporal reuse. Specifically, we merge reservoirs from neighboring pixels (spatial reuse) and previous frames (temporal reuse).

As $i$ increases, the influence of temporal reuse gradually extends to encompass contributions from all past $i$ frames. Similarly, spatial reuse progressively incorporates information from a larger region of the screen. This expansion occurs because, in each iteration, a pixel's reservoir merges with information from its neighboring pixels, and in subsequent frames, those neighboring pixels' reservoirs have already integrated data from their own surroundings. Consequently, both spatial and temporal reuse effectively propagates information across increasingly broader spatial and temporal domains.

## B  MORE EXPERIMENTAL RESULTS

### B.1  PER-SCENE QUALITATIVE RESULTS IN TENSOIR DATASET

We provide per-scene qualitative comparisons of all 4 scenes in the TensoIR synthetic dataset in Figs. 11 and 12. We compare the results of material (albedo, roughness), normal, novel view synthesis (NVS), and relighting results under 2 different environment maps per scene. Owing to our more accurate material estimation, we produce more realistic relighting results, including more accurate shadows, specular highlights and inter-reflections.

### B.2  PER-SCENE QUALITATIVE RESULTS IN OWL DATASET

We provide per-scene qualitative comparisons on the real-captured OWL dataset in Fig. 13 and Fig. 14. Our method demonstrates superior quality in the relighting appearances, while baseline methods suffer from color bias (*e.g. Tpiece* scene in Fig. 13), incorrect lighting effects (*e.g.* highlights in *Gamepad* scene in Fig. 14) or missing details (*e.g.* textures in *Antman* scene in Fig. 13). Although it is impossible to obtain ground truth material parameters in the real dataset, it can be intuitively observed that our method produces more reasonable material estimations.

### B.3  MORE ABLATION STUDIES

*Indirect illumination:* Fig. 7 illustrates the reconstructed albedo using our full model and an ablation model without indirect lighting, verifying that introducing indirect illumination can significantly improve the quality of material estimation.

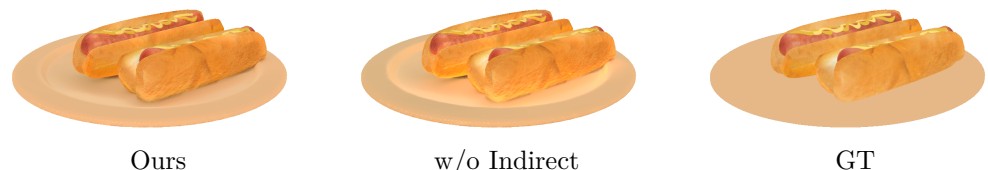

| Ours | w/o Indirect | GT |

Figure 7: **Ablation studies on indirect illumination.** Due to the lack of indirect illumination, specular reflections near the plate are heavily embedded in the reconstructed albedo. In contrast, with indirect illumination, our method achieves a cleaner albedo recovery.

*Number of SPPs:* We analyze the novel view synthesis PSNR of the *Hotdog* scene across four different configurations with varying SPPs (from 4 to 64) as shown in Fig. 8. The configurations include path tracing without indirect illumination (PT), path tracing with indirect illumination (PT full), path tracing with reservoir sampling but without indirect illumination (ReSTIR), and path tracing with both reservoir sampling and indirect illumination (ReSTIR full). The results demonstrate that the "ReSTIR full" configuration achieves the highest PSNR across all SPPs, confirming the efficacy of both reservoir sampling and indirect illumination. While the PSNR for "PT" is higher than "PT full", this does not mean the indirect illumination has a negative effect. Due to the lack of indirect illumination, "PT" bakes the specular reflection into the albedo, while "PT full" cannot capture

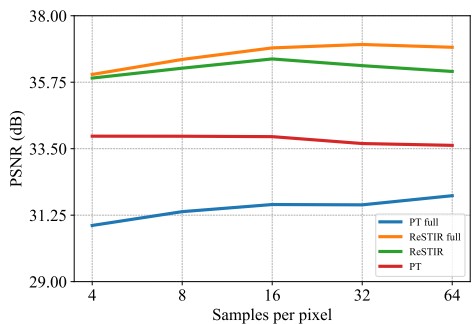

Figure 8: Novel view synthesis PSNR of *hotdog* across various SPPs.

the specular reflections due to the severe rendering noise at low SPP. As SPP increases, the specular reflections reconstructed by "PT full" become more accurate, while lower rendering variance leads to more specular ambiguity in "PT", causing a decline in PSNR. At the same time, Fig. 8 shows that a configuration with SPP higher than 32 does not necessarily improve the reconstruction accuracy. Thus, we use 32 SPP as the default configuration for all our experiments in this section.

*Neural radiance field rendering:* As described in Section 3.2, we employ two rendering methods: neural radiance field rendering and physically-based surface rendering, to jointly optimize the reconstructed geometry. Here we demonstrate the necessity of the neural radiance field rendering. As illustrated in Fig. 9, relying solely on surface rendering (the third column) leads to inaccuracies in the geometry, which subsequently affects the reconstruction of materials.

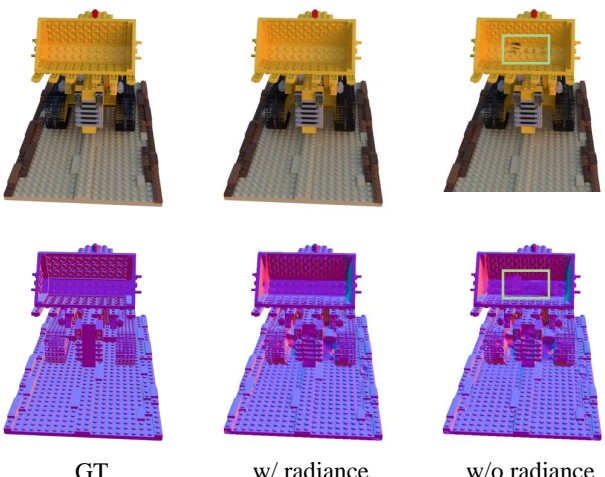

Figure 9: **Effect of neural radiance field rendering.** Top: Results of novel view synthesis. Bottom: Reconstructed normals.

## C    LIMITATIONS AND FUTURE WORK

The performance of our optimization relies on the initial coarse geometry obtained in stage 1. While NeuS2 generally reconstructs plausible geometries in most cases, the extracted mesh still contains noticeable errors, particularly in areas with high specularity or fine details. These inaccuracies can lead to incorrect material estimation and compromised novel view synthesis in the affected areas (see Fig. 10). Enhancing the geometry optimization capabilities remains an area for future research. It would also be possible to combine our

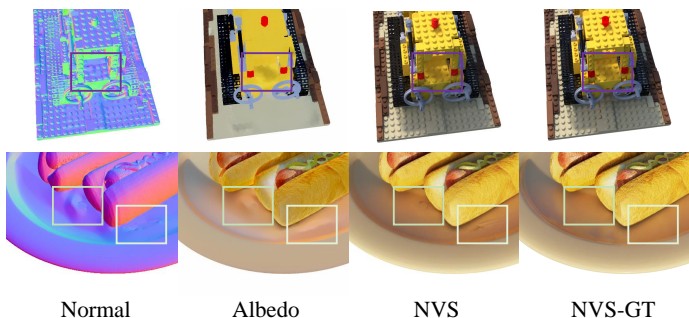

Normal      Albedo      NVS      NVS-GT

Figure 10: **Limitations: incorrect material estimation on the *NeRF Synthetic* dataset.** The primary challenges are due to high-frequency details and highly specular regions, as shown in the *Lego* scene and the *Hotdog* scene, respectively.

method with recent methods targeting specular reflections Ge et al. (2023); Wu et al. (2024); Wang et al. (2024) to achieve higher-quality reconstruction and rendering of highly specular scenes. Furthermore, our physically-based inverse rendering framework currently does not account for the gradients of non-primary rays, which could potentially improve reconstruction accuracy. However, including these gradients would significantly increase memory demands and computational costs. Exploring efficient methods for gradient computation is another avenue for future work.

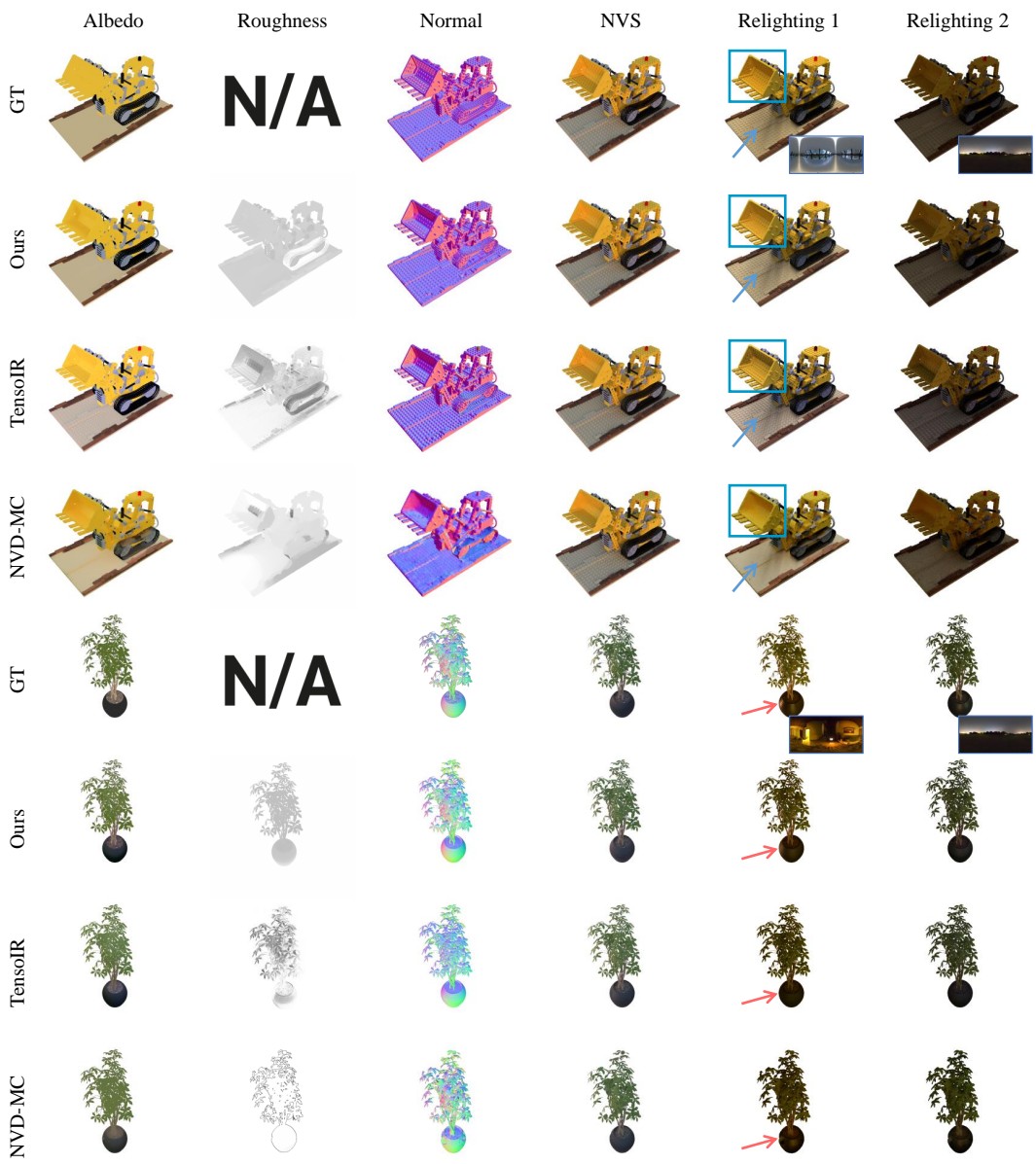

Figure 11: **Qualitative results on *Lego* and *Ficus* scenes in TensoIR dataset.** The corresponding environment map for relighting is placed on the bottom right of the GT relighting result.

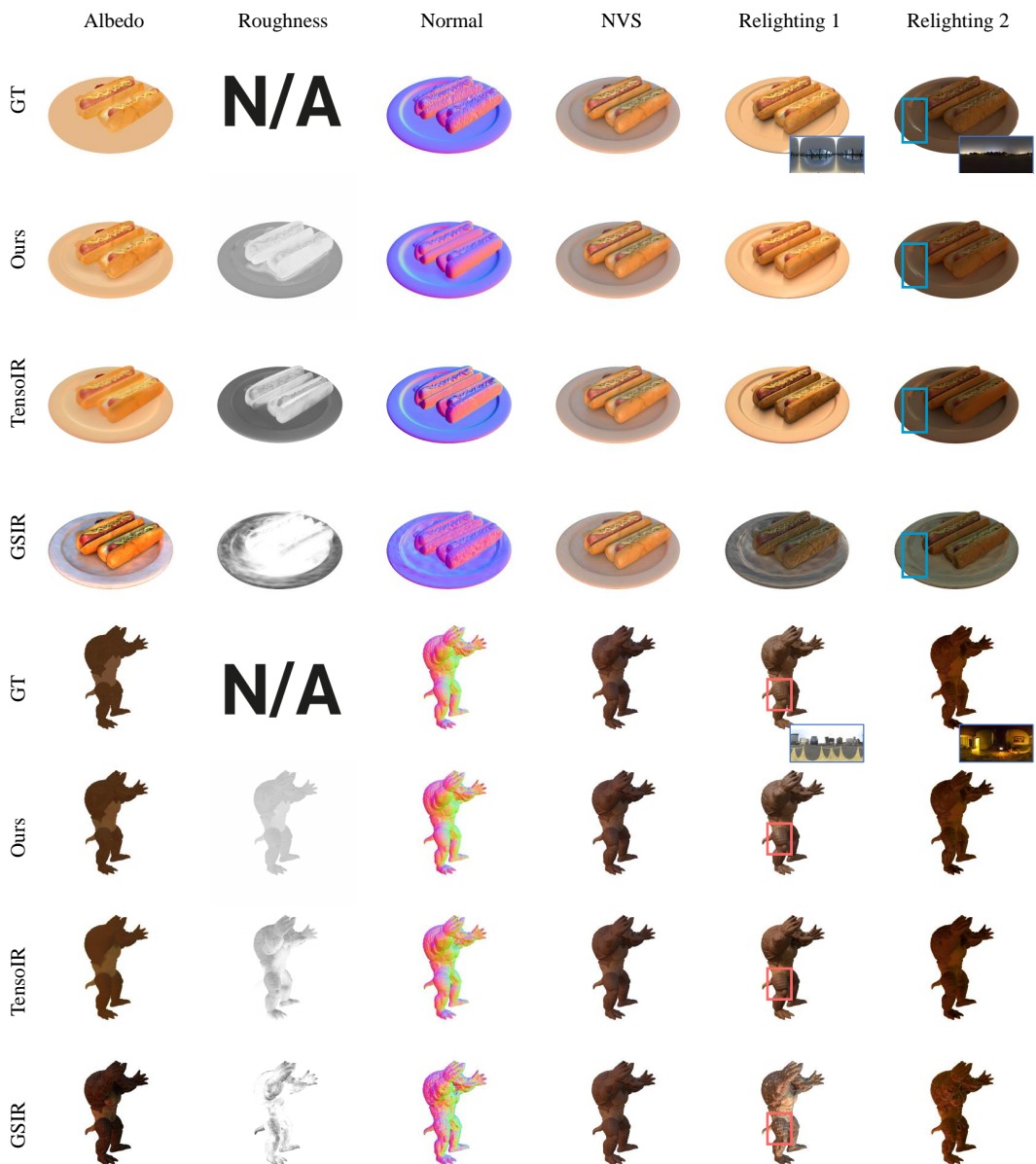

Figure 12: **Qualitative results on the *Hotdog* and *Armadillo* scenes from the TensoIR dataset.** The corresponding environment map used for relighting is displayed at the bottom right of each GT relighting result.

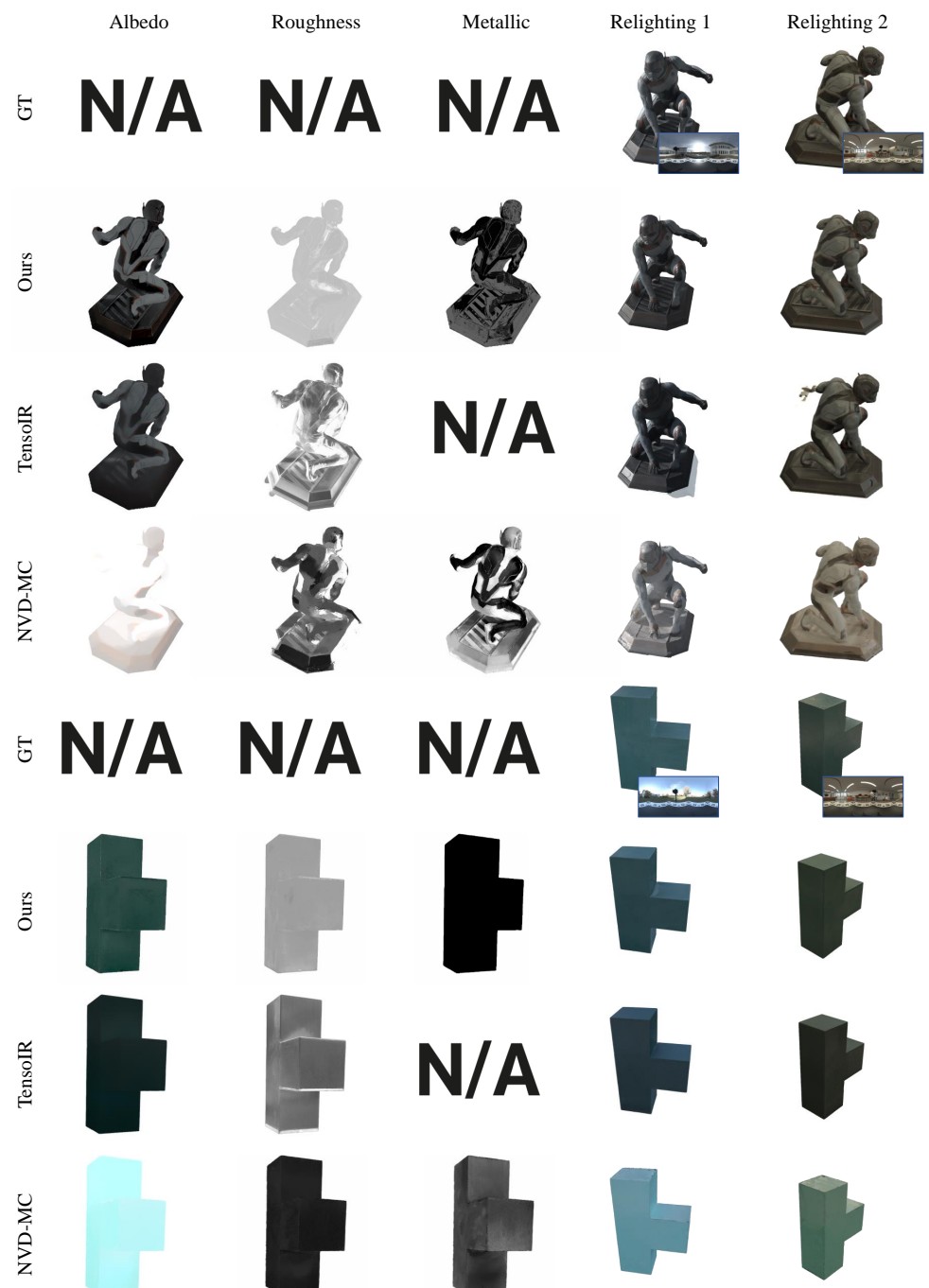

Figure 13: **Qualitative comparison on the Object-with-Lighting dataset (part 1).** Chosen from *Antman* and *Tpiece* scenes.

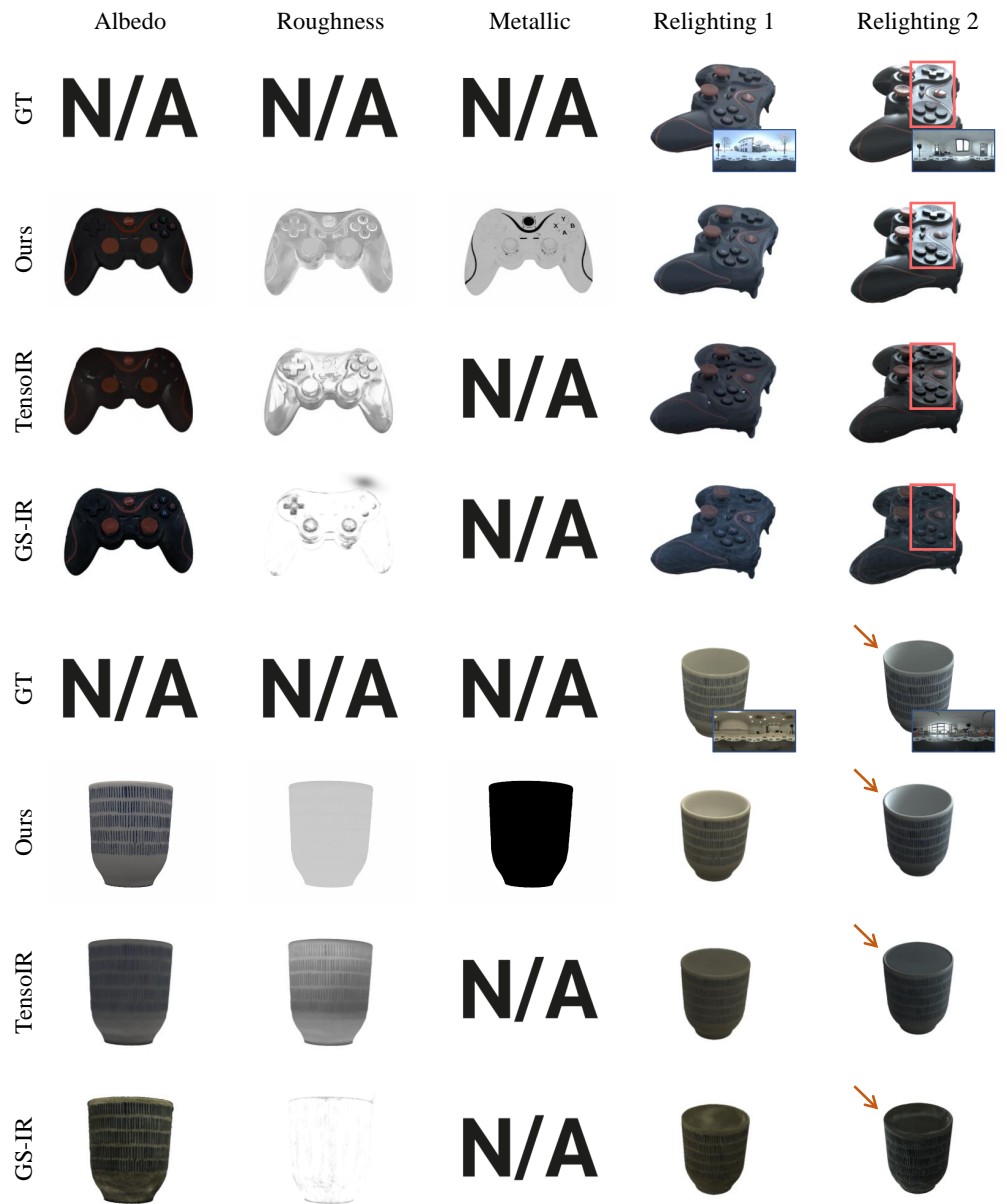

Figure 14: **Qualitative comparison on the Object-with-Lighting dataset (part 2).** Chosen from *Gamepad* and *Porcelain Mug* scenes.

