# OpenReview forum: "Inverse Rendering using Multi-Bounce Path Tracing and Reservoir Sampling"
_ICLR.cc/2025/Conference — ICLR 2025 Poster_

### Official Review · Reviewer_Kcec · 2024-10-31

**Soundness:** 3
**Presentation:** 3
**Contribution:** 3
**Rating:** 6
**Confidence:** 4

**Summary:**

The paper presents MIRReS to jointly optimize geometry, materials, and lighting. The authors propose to extract an explicit triangular mesh in the initial stage. Subsequently, they employ physically-based rendering and utilize multi-bounce path tracing and Monte Carlo integration for indirect illumination. They also incorporate a novel technique called reservoir sampling to address noise in Monte Carlo integration.

**Strengths:**

1. The paper propose to use path tracing for indirect illumination, which use PBR to compute the indirect illumination. This can be more accurate than using radiance field cache.
2. The paper propose to use reservoir sampling in the integral of rendering equation (only for direct light).

**Weaknesses:**

1. Compare to using implicit field as geometry, the use of triangle mesh may harm the rendering equality. This is common as the mesh extraction can lead to an degration in geometry quality.
2. The paper proposes to refine mesh with fixed face topology, only an offset of each vertice can be optimized, which I think the influence of the refinement is limited.
3. A derivation of Equation 9 can improve the clarity. And $p_{dir}$ in $\gamma_i$ remains unkown.

**Questions:**

see weakness

---

> ### Author Response · Authors · 2024-11-20
>
> We thank the reviewer for appreciating our contributions. Below, we address your concerns and questions in detail.
>
> > Compare to using implicit field as geometry, the use of triangle mesh may harm the rendering quality. This is common as the mesh extraction can lead to an degradation in geometry quality.
>
> We recognize the potential challenges associated with extracting meshes from implicit fields, which may compromise geometric quality. Therefore, we adopted a relatively high marching cubes resolution (256 or 512). Furthermore, the extracted mesh serves only as the initial geometry, which undergoes further optimization during the training process. This optimizing, driven by rendering loss, effectively resolves initial degradation and produces high-quality results. For example, in the Hotdog scene (Fig. 6), our method recovers more texture details compared to implicit methods such as TensoIR.
>
> In addition, explicit geometry offers advantages in ray intersection calculations. Implicit representations, such as SDFs, typically rely on sphere tracing for ray intersection, which is both less accurate and less efficient. Consequently, our mesh-based representation does not compromise rendering or geometry quality.
>
> > The paper proposes to refine mesh with fixed face topology.
>
> We acknowledge that our method does not optimize topology, and we will explicitly include this as a limitation in the revised paper. To the best of our knowledge, there are currently no effective methods for topology optimization during training. While approaches such as DMTet[*] can optimize topology, they often generate geometry with significant issues, such as holes and self-intersections. We provide a comparison of our method with DMTet and TensoIR (① and ② in [https://iamnotaproject.github.io/reb/mesh.html](https://iamnotaproject.github.io/reb/mesh.html)). It is worth noting that TensoIR's normal results directly queried from the implicit surface representation, rather than derived from an extracted mesh.
>
> > A derivation of Equation 9 can improve the clarity.
>
> We apologize for any lack of clarity in our explanation of Equation 9. Specifically, $p_{\text{dir}}(\omega_i)$ represents the distribution of the sampled ray (as defined after Eq. 8), while $\gamma_i$ denotes the sampling weight assigned to each sample during the reservoir sampling process (L312). For a more detailed description of the reservoir sampling process, please refer to the shared comment provieded earlier.
>
> [*] Shen T, Gao J, Yin K, et al. Deep marching tetrahedra: a hybrid representation for high-resolution 3d shape synthesis[J]. Advances in Neural Information Processing Systems, 2021, 34: 6087-6101.

---

> > ### Comment · Reviewer_Kcec · 2024-11-23
> >
> > I have read the authors' rebuttal. I still have some questions:
> > 1. How do the authors model the $L_{env}(\omega_i)$ in training? Using an implicit function? If so, how do the authors sample from a distribution which is proportional to $L_{env}(\omega_i)$?
> > 2. TensoIR proposes to use stratified sampling in training, and use importance sampling on ground truth environment light in relighting. But the authors propose to directly use importance sampling in training, I'm conerned about the instablility in optimization.

---

> > > ### Author Response · Authors · 2024-11-23
> > >
> > > Thanks for your reply! We'll further answer your questions as follows:
> > > 1. **Environment lighting sampling**: $L_{env}(\omega_i)$ is modeled by an explicit environment map, rather than implicit functions (see L268-269 in our paper). With all environment map values explicitly available, the PDF proportional to $L_{env}(\omega_i)$ can be directly computed and sampled. This sampling process can be computed in parallel and, therefore, is efficient.
> > > 2. **Importance sampling VS stratified sampling**: Thanks for this valuable perspective! We acknowledge that importance sampling and stratified sampling have both pros and cons: stratified sampling does not suffer from noises but has a low capability of capturing high-frequency rendering details such as specular reflections and highlights, while importance sampling is vice versa. However, our reservoir sampling can greatly reduce the noise level compared to importance sampling (like in NVD-MC), thus significantly overcoming the main weakness of importance sampling while retaining the advantages over stratified sampling. We believe that the reduced noise level is sufficient for stable optimization, and experimental comparisons have demonstrated that our method achieves the best quality.

---

### Official Review · Reviewer_fqgJ · 2024-11-02

**Soundness:** 3
**Presentation:** 4
**Contribution:** 3
**Rating:** 8
**Confidence:** 4

**Summary:**

The novel mesh-based reconstruction method, Multi-bounce Inverse Rendering using Reservoir Sampling (MIRReS), introduced in this paper, decomposes geometry, material parameters (diffuse albedo and roughness), and illumination from multi-view images using multi-bounce path tracing with Monte Carlo integration.
The method produces reconstruction results of unparalleled quality due to the authors' use of global illumination (Monte Carlo ray tracing) to explicitly and physically account for indirect lighting. This is only possible because, for the first time in Inverse Rendering, ReSTIR (Reservoir-based Spatio-Temporal Importance Resampling) is employed to accelerate the costly global rendering.
An evaluation of the new method and a comparison with the latest and best reconstruction methods for geometry, material and lighting (NVDiffRecMC 2022, TensorIR 2023 and GS-IR 2023) shows the comparatively very good reconstruction quality, both of geometry and albedo and with regard to relighting the reconstructed scenes.

**Strengths:**

1.) The paper is very well written and easy to follow.
2.) It contains a very good summary of the current state of the art on Inverse Rendering for Shape, Light and Material reconstruction.
3.) The idea to use Reservoir-based Spatio-Temporal Importance Resampling to reduce the computational load of a global illumination calculation is novel and should definitely be published.

**Weaknesses:**

There is no code available to evaluate the method  independently.

The evaluation and the (very good) comparison with other state-of-the-art methods lack a comparison of the results of the specular reflection. Is this estimate much worse than for the other methods? In the appendix, this is mentioned as a limitation of the procedure.

**Questions:**

As there is currently no code from which further details about the method can be obtained. This leaves a few questions unanswered.
1.) How does the rough estimate of the geometry in stage 1 of the algorithm affect the rendering, and how does the algorithm determine when the geometry is sufficiently accurate to begin material reconstruction and global illumination simulation in stage 2?
2.) In this context, it would be interesting to know how the inaccurate initial estimate of the geometry affects the estimate of the specular reflections.
3.) The paper mentions an important feature: the reconstruction result as a triangle mesh is compatible with current graphics engines and CAD software. However, the texture is modeled implicitly using Instant NGP in the method. How is it transferred to the triangle mesh?

---

> ### Author Response · Authors · 2024-11-20
>
> We thank the reviewer for appreciating our contributions and results. Below, we address your concerns and questions in details.
>
> > There is no code available to evaluate the method independently.
>
> We will make our source code publicly available upon acceptance of the paper.
>
> > ... how does the algorithm determine when the geometry is sufficiently accurate...
>
> We use the default configuration of NeuS2, as it allows the model to converge effectively and produce satisfactory results. To investigate the impact of training steps in their configuration, we conducted experiments by reducing the steps to 10% and 50% of the default configuration. The results showed a gradual improvement in performance as the number of steps increased. However, since the default configuration already delivers sufficient accuracy and quality, we determined that additional adjustments to the steps are unnecessary.
>
> | Dataset   | nvs     | albedo  | relight |
> |-----------|---------|---------|---------|
> | lego_10%  | 31.8439 | 25.8862 | 29.6523 |
> | lego_50%  | 32.547  | 26.262  | 30.0056 |
> | full      | 34.39   | 28.02   | 30.827  |
>
> > ... lack a comparison of the results of the specular reflection. Is this estimate much worse than for the other methods? ...
>
> Our method is not restricted to diffuse materials and is capable of handling mildly rough specular reflections. However, we acknowledge as a limitation that it cannot handle extremely sharp mirror reflections (e.g. the ShinyBlender dataset or the NeRO dataset).
>
> We already provide comparisons with baselines in scenes featuring specular reflections: (1) In the relighting result of Hotdog (Fig. 6), our method produces better specular highlights on the edge of the disk compared to baseline methods; (2) In the “Relighting 2” result of Gamepad (Fig. 14 in the Appendix), we specifically selected a view angle with strong specular reflections on the gamepad. Our method faithfully recovers these reflections, while baseline methods fail.
>
> > In this context, it would be interesting to know how the inaccurate initial estimate of the geometry affects the estimate of the specular reflections.
>
> Here, we illustrate how an inaccurate initial geometry estimate impacts the estimation of specular reflections in this example ([https://iamnotaproject.github.io/reb/specular.html](https://iamnotaproject.github.io/reb/specular.html))
>
> > ... the texture is modeled implicitly using Instant NGP in the method. How is it transferred to the triangle mesh?
>
> Although material properties are implicitly modeled by Instant-NGP during the training stage, texture maps can be extracted during inference using the well-defined tool: [xatlas: https://github.com/jpcy/xatlas](https://github.com/jpcy/xatlas) . Thank you for highlighting this point and we will include a detailed explanation in the Appendix.

---

> ### Author Response · Authors · 2024-11-25
>
> Dear reviewer,
>
> We would like to express our sincere gratitude for the time and effort you have dedicated to reviewing our work and providing valuable feedback. As the deadline for discussion is approaching on November 26, we kindly ask whether our response has sufficiently addressed your concerns.

---

### Official Review · Reviewer_QNEc · 2024-11-03

**Soundness:** 2
**Presentation:** 2
**Contribution:** 1
**Rating:** 3
**Confidence:** 4

**Summary:**

This paper proposes a framework for joint reconstruction of shape, material and lighting. The framework consists of two stages. In the first stage, it uses neural SDF optimization to reconstruction the initial object geometry and radiance field. In the second stage, it uses differentiable rendering combined with differentiable marching cube to optimize geometry, materials and lighting, with indirect illumination being properly handled. Experiments on synthetic and real object datasets show that the proposed framework achieves compelling results compared to several strong baselines.

**Strengths:**

1. A novel inverse rendering framework that combines the advantages of neural SDF reconstruction and physically-based differentiable rendering.
2. The paper overall is easy to follow and well organized.
3. High-quality inverse rendering results on both synthetic and real datasets compared to several prior strong baselines.

**Weaknesses:**

1. Technical novelties of the paper is very limited.

This paper's major contributions are covered by two previous works that are not properly cited. Neural-PBIR, Cheng et al.. ICCV 2023 adopts a similar pipeline to first train a neural SDF to get high-quality geometry and then run PBDR to further optimize geometry, materials and lighting. Neural-PBIR also models multiple bounces and uses unbiased gradient for geometry optimization in the PBDR stage. "Parameter-space ReSTIR for Differentiable and Inverse Rendering", Chang et al., SIGGRAPH 2023 adopts Reservoir sampling into inverse rendering by exploring the temporal coherence across gradient iterations. It provides a much deeper theoretical analysis and more detailed comparisons.

It will significantly enhance the paper if we can explicitly discuss how the proposed method differs from or improves upon Neural-PBIR and Parameter-space ReSTIR. We may consider adding a paragraph comparing the propoesd approach to these prior works, highlighting any key differences or advancements.

2. Design choices of Reservoir sampling may not be completely technically sound.

It is reasonable to explore temporal coherence to reduce the noise and hence reduce the sample. as demonstrated in Chang et al. SIGGRAPH 2023. However, the motivations described in the Eq. (9) emphasizes that there is no analytical integral of $L(\omega_i)f(\omega_i, \omega_o)$ so that we need reservoir sampling to approximate the distribution of $L(\omega_i)$. This makes much less sense as there is a clear solution to sample the environment map lighting and we can even consider MIS to sample both BRDF and lighting. The last paragraph of Section 4.1 mentioned temporal reuse and spatial reuse but we may need more detailed discussion.

It will be beneficial to have a deeper discussion to why we need reservoir sampling to sample environment map and also more analysis on the impact of temporal reuse and spatial reuse.

3. Optimization time is too long.

The total optimization time is 4.5 hours, which is 4 times longer than Neural-PBIR. This is contradictory to the motivation of using reservoir sampling to reduce the number of sample. One thing to note is that the running speed of PBDR heavily depends on system engineering. This paper may consider using more advanced PBDR framework, such as Mitsuba 3, to see if the optimization time can be significantly reduced. A breakdown of the current optimization will also help us identify the issue.

4. Other minor issues:
1. Line 196 - 202: Why not directly use Neus2 for shape reconstruction? Why run instant-NGP first? Please provide more discussion and analysis.
2. Line 242: the notation $\Delta \mathbf{v} $ is in consistent with the notation in Figure 2 $\Delta \mathbf{x}$. Please ensure notation consistency to avoid confusion.
3. Line 263: Why skip metallic? Please provide reasoning on this design choice and the potential impacts.
4. Experiments: Can we include experiments on Stanford-ORB to have a more complete comparison with prior methods? Can we show visualization of the environment maps to help reader understand the lighting reconstruction quality?

**Questions:**

I would like to see more arguments about technical novelties in the paper and how this paper is different from prior work. In addition, more technical details of reservoir sampling will be welcome.

---

> ### Author Response · Authors · 2024-11-20
>
> We thank the reviewer for your appreciation of our contribution and results, and we'll address your concerns and questions below.
>
> > ...explicitly discuss how the proposed method differs from or improves upon Neural-PBIR and Parameter-space ReSTIR...
>
> Our work differs from the two mentioned methods in the following ways:
>
> 1. **Neural-PBIR**: The Neural-PBIR framework consists of three components: (1) geometry extraction, (2) material and lighting distillation, and (3) further optimization of geometry and material using an off-the-shelf PBIR renderer [1]. The core of this inverse rendering process is initializing high-quality geometry and materials using the optimized NeuS network and learnable per-vertex material parameters. The final optimization using an existing PBIR renderer can be considered as an independent post-processing step. In fact, all inverse rendering methods that utilize explicit geometry and material texture representations, including ours, can integrate this PBIR renderer for refinement. Thus, our approach corresponds to the first two components of Neural-PBIR. While the geometry extraction methods in Neural-PBIR and our approach are similar (both train an SDF network and extract explicit geomeetry via the marching cubes algorithm), there are **substantial differences in material estimation**:
>
> In terms of **material representation**, Neural-PBIR assigns per-vertex albedo and roughness values for explicit geometry and uses linear interpolation for triangle faces. In contrast, our method employs Instant-NGP MLPs to represent materials at any point in space, and additionally accounting for metallic. Although MLP-based representations have a slower optimization speed, they offer resolution independence, and are not constrained by the number of vertices or fixed mesh topology, and allow flexible optimization with richer material details.
>
> For **indirect illumination**, Neural-PBIR adopts a strategy similar to TensoIR [2], which queries the neural radiance field for indirect ray radiance.  This approach heavily depends on the radiance field quality, making it prone to error. In contrast, our method computes indirect illumination directly through multi-bounce path tracing, achieving more accurate results by avoiding reliance on potentially suboptimal radiance fields.
>
> During the **rendering** process, Neural-PBIR employs *stratified* sampling of 256 fixed light directions per vertex to compute color. While this precomputed sampling accelerates rendering, it can introduce errors in visibility and indirect illumination, especially if the geometry is inaccurate. In contrast, our method leverages an advanced *ReSTIR* sampling strategy, which effectively controls color variance and reduces the required sample count.
>
> Furthermore, Neural-PBIR's precomputed light directions remain fixed since the geometry does not update during the second stage. While this approach speeds up rendering, inaccuracies in geometry exacerbate errors in visibility and indirect illumination precomputations, which may affect rendering quality.
>
>  As Neural-PBIR has not released its code, we compared the albedo of the Hotdog scene as presented in their paper. We verified that their training data matches ours ([① in https://iamnotaproject.github.io/reb/MIS.html](https://iamnotaproject.github.io/reb/MIS.html)). Neural-PBIR also demonstrates high-quality environment map results in their paper by distilling the environment map from background observations in RGB image (see Eq. 12, and Appendix Sec. B.2 and Fig. 11 in their paper). This approach can be integrated as a plug-and-play module. However, our method, along with other baselines (TensoIR, NVdiffrec-MC, GS-IR), assumes only foreground observations, without leveraging this setting.
>
> [1] Zhang C, Miller B, Yan K, et al. Path-space differentiable rendering[J]. ACM Transactions on Graphics (TOG), 2020, 39(4): 143: 1-143: 19.
>
> [2] Jin H, Liu I, Xu P, et al. Tensoir: Tensorial inverse rendering[C]. Proceedings of the IEEE/CVF Conference on Computer Vision and Pattern Recognition, 2023: 165-174.

---

> > ### Comment · Reviewer_QNEc · 2024-11-26
> > **Discussion on comparisons with Neural-PBIR**
> >
> > Thanks authors for the detailed response and new experimental results!
> >
> > It is probably not fair to compare the propose method with only the first 2 stages of an existing method and draw the conclusion that the proposed method improves the existing method. Besides, the second stage of the prior method, as mentioned in the paper, is to efficiently get a rough material and lighting estimation for initialization. Therefore, the two methods should be compared together as a whole.
> >
> > **Material representation** In terms of material representation, prior method eventually optimizes a high resolution texture map which is more detailed compared to per-vertex color (as shown in Fig. 7, left). While MLP can achieve infinite high resolution reconstruction, in practice it is also limited by network capacity and computational cost. Therefore, it may not always reconstruction high-quality details. For example, in the prior work,  the texture map extracted from MLP directly has less details compared to the texture map optimized per texel as shown in Fig. 7 left. Experiments will greatly help to answer which representation is better. I would therefore suggest testing the proposed method on Stanford-ORB to have a direct comparison.
> >
> > **Indirect illumination** Neural-PBIR only uses neural radiance field to model indirect illumination in the second stage of distillation, which provide a good initialization. In the third stage, Neural-PBIR uses physically-based differentiable rendering (PBDR) that models indirect illumination to optimize the material parameters and shows improvements in solving color bleeding issue, as shown in Fig. 7. The idea of using PBDR is the same as the proposed method while the optimization process is faster.
> >
> > **Rendering** This only covers the first two stage of Neural-PBIR, whose purpose is to get accurate geometry and a rough initial material and lighting estimation to accelerate the third stage optimization. And the third stage models the full indirect illumination and lighting similar to the proposed method.
> >
> > The hotdog example looks great and is very impressive. In the meantime, a more thorough comparison on existing benchmark will also be useful to answer many questions. I will really appreciate it if we can include the comparisons on Stanford-ORB benchmark to better understand the advantages of the proposed method.

---

> ### Author Response · Authors · 2024-11-20
>
> 2. **Parameter-space ReSTIR**: We’ll describe the differences between our method and Parameter-space ReSTIR (Chang et al. 2023, abbreviated as “P-RIS” in the following) below:
>     1. Different settings: P-RIS assumes known geometry (mesh) and focuses on optimizing texture map parameters on the mesh, while our method tackes a general inverse rendering task that jointly optimizes geometry, material, and lighting.
>     2. Different motivation: P-RIS aims to simplify gradients in inverse rendering by transitioning from memory-intensive pixel space to parameter (texel) space and leverages reservoir sampling to efficiently estimate gradients. In contrast, our method uses reservoir sampling to mitigate Monte Carlo noise in path tracing, enabling more robust rendering results to facilitate training convergence.
>     3. Different solution spaces of reservoir sampling: P-RIS performs reservoir sampling in texture space to estimate gradients, whereas our method applies reservoir sampling in primary space to estimate rendering results (first bounce of path tracing).
>
> 	Although both methods leverage reservoir sampling, they apply it to fundamentally different scenarios.
>
> > … the motivations described in the Eq. (9) emphasizes that there is no analytical integral of $L(\omega_i)f_r(\omega_i,\omega_o)$ so that we need reservoir sampling to approximate the distribution of $L(\omega_i)$. This makes much less sense as there is a clear solution to sample the environment map lighting and we can even consider MIS to sample both BRDF and lighting…
>
> We believe there has been a misunderstanding. Our reservoir sampling is used to approximate the distribution of $L(\omega_i)f_r(\omega_i,\omega_o)$, not the distribution of $L(\omega_i)$. Therefore, our method is different from MIS sampling, which only combines the two samples obtained from BRDF sampling and light sampling using MIS weight but does not directly sample from the distribution of their product.
>
> We acknowledge that the description of reservoir sampling in our manuscript could be clearer. Please refer to our common comment provided above for a revised description of the reservoir sampling process. We will also revise the explanation in Sec. 4.1. accordingly. Again, we apologize for the lack of clarity and the confusion it caused.
>
> > It will be beneficial to have a deeper discussion to why we need reservoir sampling to sample environment map and also more analysis on the impact of temporal reuse and spatial reuse.
>
> To address this, we have included qualitative comparisons between our reservoir sampling and MIS, as well as an ablation study on the usage of spatiotemporal reuse ([https://iamnotaproject.github.io/reb/MIS.html](https://iamnotaproject.github.io/reb/MIS.html), ② and ③). These results demonstrate that our reservoir sampling strategy and spatiotemporal reuse significantly reduce noise levels, enhancing overall performance.
>
> > The total optimization time is 4.5 hours... This paper may consider using more advanced PBDR framework, such as Mitsuba 3...
>
> In addition to the vertex-based material representation and precomputed indirect illumination, Neural-PBIR directly leverages a highly optimized off-the-shelf renderer([PSDR-jit](https://github.com/andyyankai/psdr-jit)) as their PBDR framework, and similarly, Mitsuba 3 leverages [Dr.jit](https://github.com/mitsuba-renderer/drjit). We agree with your point of view that these JIT (Just-In-Time-Compiler) frameworks use heavy engineering optimization techniques to improve the performance. However, a significant restriction of using these JIT frameworks is that they only support (a very limited) pre-defined operations inside the loop. Our method uses neural networks (using PyTorch) and Instant-NGP, which are unfortunately not able to be traced by the JIT.
> In Mitsuba 3, if unsupported operations are included within the loop, the JIT optimization must be disabled to ensure proper updates to the PyTorch network. As a result, the performance will drop significantly, reverting to the behavior of a standard loop.
>
> Despite this, we believe that the 4.5-hour optimization time is acceptable, especially since it is comparable to TensoIR (reported as 5 hours in their paper). The slightly longer runtime is a worthwhile trade-off for the significant improvements in result quality delivered by our method.

---

> ### Author Response · Authors · 2024-11-20
>
> > Why run instant-NGP first?
>
> We use Instant-NGP to initialize the radiance field, facilitating the subsequent radiance field rendering (Eq. 3). The choice of Instant-NGP over NeuS2 is due to its superior novel view synthesis performance, which enhances the accuracy of radience field rendering. Please refer to Fig. 9 in the Appendix, which demonstrates the effectiveness of our radiance field rendering design.
>
> > Why skip metallic?
>
> We apologize for the misleading representation in Fig. 2, which does not include metallic. In fact, our method does support metallic optimization, and we optimize metallic for scenes in the OWL dataset (L494-495, Figs. 13 and 14 in the Appendix).
>
> For the TensoIR synthetic dataset, we skip metallic optimization, consistent with the experimental settings of prior works, including TensoIR and GS-IR. This synthetic dataset is rendered using dielectric Disney BRDF materials without metallic properties, so optimizing metallic for this dataset is unnecessary.
>
> For real-world datasets with more complex materials (e.g., OWL), we include metallic in our optimization process to achieve better quality results.

---

> ### Author Response · Authors · 2024-11-25
>
> Dear reviewer,
>
> We would like to express our sincere gratitude for the time and effort you have dedicated to reviewing our work and providing valuable feedback. As the deadline for discussion is approaching on November 26, we kindly ask whether our response has sufficiently addressed your concerns.

---

> ### Comment · Reviewer_QNEc · 2024-11-27
> **Discussion on reservoir sampling**
>
> Thanks authors for updating the text and including the extra description of RIS and ReSTIR! It helps clarify some questions of the proposed method. I will list my remaining questions below.
>
> **Technical novelty**: RIS is a well-known sampling technique in rendering. Reservoir sampling as well as spatiotemporal reuse were proposed in Bitterli et al. to reduce rendering cost, especially for the case with many light sources. Adopting these sampling technologies may reduce the optimization time and noise, but the technical novelty may not be very strong unless the quality or speed are significantly better than prior works. I would suggest more experimental results on more public available benchmark to enhance the paper, such as Stanford-ORB dataset.
>
> **Difference from parameter-space ReSTIR**: I agree that (1) is a major difference but then the following up question is we need to verify if the joint optimization can help us improve the reconstruction quality, especially geometry reconstruction quality. That will also further differ the proposed method with Neural-PBIR as in that paper differentiable rendering does not significantly improve the geometry reconstruction quality. (2) and (3) are less convincing to me because I feel P-RIS is solving a more fundamental issue of PBDR to reduce the variance of gradients while the proposed method is using reservoir sampling to reduce the noise of the rendering results, which was analyzed in prior ReSTIR paper already.
>
> **Comparisons with MIS**: Thanks a lot for preparing the results and more details of the comparisons will be really appreciated. My first question is the MIS rendering seems to have a global color shift compared to the ground-truth, which is unexpected as MIS should be unbiased. Second, how many samples are used to create the MIS and reservoir sampling results? One benefit of reservoir sampling is that we can significantly increase the value of m and reduce the value of N but this can also be unfair since creating samples can cost time. In this case, maybe total rendering time is a fairer comparison?
>
> **Optimization time**: Thanks for the analysis. If incorporating pytorch network will cause the optimization time to be significantly longer, one experiment to prove that it is worth the cost is to show that optimizing these network parameters will lead to higher reconstruction quality. Maybe it will improve the geometry quality. Maybe it will solve lighting baking issue. Such an experiment will significantly help the paper, in my opinion

---

> > ### Author Response · Authors · 2024-12-02
> >
> > We thank the reviewer for the response, and we'll address your concerns and questions below.
> >
> > ### Technical Contribution
> >
> > We argue that our method is a holistic inverse rendering framework, and our main contribution lies in the combination of all components (reservoir sampling, multi-bounce path tracing, etc.) rather than reservoir sampling alone. The key intuition of our paper is that multi-bounce path tracing provides more accurate global illumination to improve the inverse rendering quality while suffering from high variance, and on the other hand, reservoir sampling can significantly reduce the Monte Carlo variance. Based on this observation, we design our holistic framework in which both multi-bounce path tracing and reservoir sampling complement each other and are indispensable, and achieve significantly higher-quality results than baselines. We believe our main technical contribution lies in this rather than simply adopting reservoir sampling individually.
> >
> > ### Difference from Parameter-space ReSTIR
> >
> > We agree that P-RIS's derivation of gradient reparameterization is very insightful and meaningful, however, we still want to emphasize that our paper works on a completely different setting from P-RIS, as described in our previous reply. Specifically, P-RIS is based on the setting of a texture space while all other parameters and conditions are fixed, which is a relatively simple parameter space, enabling an analytical derivation of gradient reparameterization. On the other hand, our setting, which is jointly optimizing geometry, material, and lighting, is more complex, so P-RIS's derivation cannot be trivially migrated and applied and needs to be completely re-derived from scratch. We appreciate the reviewer's comment that provides an insightful viewpoint on the inverse rendering task, but we believe that this is out of the scope of this work and we can add a discussion paragraph about it as a future work.
> >
> > ### Material Representation and Optimization Time
> >
> > We acknowledge that the MLP representation is limited by network capacity, but hybrid representations like Instant-NGP have already been widely proven the ability to express high-frequency details, also validated by our albedo results.
> >
> > > in the prior work, the texture map extracted from MLP directly has less details compared to the texture map optimized per texel as shown in Fig. 7 left
> >
> > We analyze that this is because (i) they model indirect illumination using radiance field MLP predictions, which is inaccurate as analyzed in L345-358 of our paper (ii) they use stratified sampling in their second stage for rendering, which has a low capacity of capturing high-frequency rendering details such as specular reflections and highlights. On the other hand, their third stage benefits from the PBDR path tracer which improves their texture map details.
> >
> > Both leveraging PBDR, we demonstrate that the material textures reconstructed our method contain more details than Neural-PBIR in the MII comparisons. We believe that this validates the effectiveness of our MLP-based material representation scheme. Also, we believe that it is worth the cost of our slightly longer but acceptable optimization time for the higher-quality results.
> >
> > ### Comparisons with MIS
> >
> > We agree that MIS is an unbiased rendering algorithm. Regarding the seemingly global color shift observed in the 1spp result, we believe this is due to the high level of noise, which impacts the color perception of the human eye. This phenomenon can also be observed in other rendering papers(e.g., ReSTIR-related works)

---

> ### Comment · Reviewer_QNEc · 2024-11-27
> **Summary**
>
> Thank authors for the response and it partially solves my questions! I think the following two experiments may improve the paper.
>
> 1. Experiments on Stanford-ORB dataset. This will answer questions about comparisons with Neural-PBIR and if the the proposed framework improves prior published works.
>
> 2. More experiments demonstrates the benefit of connecting neural representation and physically-based differentiable rendering. It will be great if that can lead to better geometry reconstruction or less color bleeding artifacts. This will justify the design choice of not using the most efficient optimization framework and also helps distinguish from prior works.
>
> Also, I may still suggest that directly adopting ReSTIR in inverse rendering may not be considered as a major contribution and the major contribution should still be an inverse rendering system that achieves better reconstruction quality. The reason is because any sampling methods that reduce variance can be used in inverse rendering task to accelerate the optimization process. Unless it is specifically designed for inverse rendering, it does not bring too much novel insights. Also the evaluation needs to be more comprehensive as different sampling methods may work very differently for different lighting lighting and scene configurations.

---

> > ### Author Response · Authors · 2024-12-02
> >
> > Following your suggestion, we evaluated on the Stanford-ORB dataset and compare it with Neural-PBIR. However, the complete Stanford-ORB dataset requires training on 42 scenes, and given the limited discussion time, we believe it is not feasible to conduct such a large-scale experiment within the rebuttal period. Therefore, we selected the scenes shown in Fig. 10 of the Neural-PBIR paper for our experiments and compare our results with theirs. We hope you understand the constraints under which we are working.
> >
> > Please refer to [https://iamnotaproject.github.io/reb/pbir.html](https://iamnotaproject.github.io/reb/pbir.html) for per-scene comparison, where Neural-PBIR's results are directly obtained from the screenshot of their paper since their code has not been open-sourced. Our method produces higher-fidelity relighting results, including more accurate color tone, shadows and highlights.
> >
> > Additionally, we also evaluated on MII dataset, another dataset which Neural-PBIR experiments on, and compare the reconstructed albedo. Our method is able to recover high-frequency texture details in the albedo while eliminating baked-shadow artifacts.

---

### Official Review · Reviewer_Cqns · 2024-11-04

**Soundness:** 3
**Presentation:** 4
**Contribution:** 3
**Rating:** 8
**Confidence:** 3

**Summary:**

This paper proposes a two-stage inverse rendering framework to jointly reconstruct geometry, materials, and lighting from multi-view images. In this framework, geometries are represented as triangular meshes with trainable vertex offsets, and this representation enables explicit path tracing technique similar to the classic PBR pipeline, which is sped up with reservoir sampling for direct illumination. With these contributions, the reported results showed better reconstruction quality for geometry, materials, and lighting both quantitatively and qualitatively.

**Strengths:**

1. This work adapts well-studied PBR techniques, such as path tracing and reservior sampling, to the domain of inverse rendering. I personally appreciate the adaptation of these computer graphics knowledge to address a more computer vision problem, and the results do show an improvement with these contributions.
2. The most related work to this submission is NVdiffrec-MC. From the reported results, this work achieves a better performance both quantitatively and qualitatively.
3. This paper is well written and easy to follow. The entire formulation is physically sound and all the theoretical details are explained in an intuitive way for readers with background in physics-based rendering and inverse rendering. Sufficient ablation studies are provided to verfiy the effectiveness of each component.
4. Given the long-standing and inherently ill-posed nature of inverse rendering, I believe this work makes a valuable contribution to this field. It can inspire further research and solutions aimed at advancing the understanding and solving of this challenging problem.

**Weaknesses:**

1. This work relies on a high-quality intialization of geometry, for which the authors chose NeuS2. However, the authors acknowledged that some artifacts may still persist in certain areas, which cannot be fully corrected by the subsequent refinement step. Additionally, they discussed the instability of DMTet for geometry representation, which is used by NVdiffrec-MC. I would be interested to see some visual comparisons between these two geometry representations.
2. As the formulation only considers primary rays, it cannot handle objects with transparent materials (e.g., a glass sphere exhibiting Fresnel effects). This might be an interesting direction to explore for future work.

**Questions:**

1. In Lines 084 -- 086, you mention the effectiveness of a denoiser inspired by NVdiffrec-MC; however, I was unable to find further discussion or detailed follow-up content about it in the manuscript. Could you elaborate on the design of this denoiser and how effective it is for the final rendering?
2. Since an explicit environment map is also being optimized, it would be helpful to include visualizations of the optimized environment maps for some scenes to better understand the impact of this optimization.
3. In Lines 401 -- 403, you mentioned some adjustments made to the albedo and rendered images. Could you provide more details about the specific adjustments used?

---

> ### Author Response · Authors · 2024-11-20
>
> We sincerely thank the reviewer for appreciating our contributions and results. We agree that computer graphics and computer vision can complement and enhance each other. Below, we address your concerns and questions in details.
>
> > comparisons with DMTet
>
> DMTet optimizes geometry starting from a randomized initial guess and allows for topology changes during the optimization process. However, it often generates geometry with issues such as holes and self-intersections. Our goal in using this optimization method is to achieve more stable geometry. We provide comparisons between our geometry optimization results and those of DMTet, along with a comparison of our mesh before and after optimization (① and ② at the annoymous GitHub page [https://iamnotaproject.github.io/reb/mesh_denosier.html](https://iamnotaproject.github.io/reb/mesh_denosier.html)).
>
> > it cannot handle objects with transparent materials
>
> We acknowledge that our method currently cannot handle transparent objects. Following your suggestion, we will include a discussion of this limitation in our revised paper and propose it a direction for future work.
>
> > Could you elaborate on the design of this denoiser?
>
> The denoiser used in our pipeline is an off-the-shelf tool that we directly incorporate into method. We apologize for not providing a detailed explanation of its design in the original paper. Please refer to another comment for the details of our denoiser. We'll include a detailed elaboration on the denoiser as a new paragraph in the Appendix of our revised paper.
> To demonstrate its effectiveness, we also present rendering results with and without the denoiser in the annonymous GitHub page ([https://iamnotaproject.github.io/reb/mesh_denosier.html](https://iamnotaproject.github.io/reb/mesh_denosier.html), ③).
>
> > visualizations of the optimized environment maps
>
> Existing inverse rendering methods generally utilize two types of representations for environment lighting: spherical Gaussians (SG) (e.g., in TensoIR) or trainable environment maps (as used in our method). Our decision to use trainable environment maps is based on the observation that SG representations often struggle to capture high-frequency details. Besides, TensoIR adopts a resolution of 16×32 for computational efficiency, whereas we use a higher resolution of 128×256, enabling us to represent high-frequency details more effectively. This is consistent with the results achieved by our method.
>
> We acknowledge that recovering environment maps is a highly ill-posed problem, since input images primarily provide low-frequency directional intensity information (from highlights and shadows) but lack high-frequency details. Consequently, the optimization gradient is sparse and noisy. As shown in ([https://iamnotaproject.github.io/reb/mesh_denosier.html](https://iamnotaproject.github.io/reb/mesh_denosier.html), ④), our optimized environment map accurately captures directional lighting distribution but is inevitably noisy due to the inherent nature of Monte Carlo methods. Despite this, we note that environment map reconstruction is not the primary focus of our approach. The main objective of our inverse rendering method is to utilize the reconstructed geometry and material for downstream applications such as relighting and rendering, where the optimized environment map is ultimately discarded. Despite the noise in the environment map, our method achieves significantly higher-quality geometry and material reconstructions compared to baseline methods.
>
> > In Lines 401 -- 403, you mentioned some adjustments made to the albedo...
>
> For albedo visualization, we follow the convention established by TensoIR (which, in turn, references NeRFactor). This approach is based on the assumption that recovering the absolute brightness of the albedo and illumination from an image is inherently ambiguous. For instance, multiplying the albedo by 2 while simultaneously halving the environment lighting produces the same rendering result. To address this ambiguity, we perform a least-square alignment between the brightness of our reconstructed albedo and the GT albedo. Specifically, each RGB channel of the albedo is scaled by a global scalar factor that minimizes the mean squared error w.r.t. the GT albedo.
> For real images, where GT albedo is not available, we apply a similar least-square brightness scaling step to the relighted images. It is important to note that this is a fair comparison, since we apply this adjustment across all baselines.

---

> ### Author Response · Authors · 2024-11-20
>
> ### Descriptions of Denoiser
>
> We adopt the Edge-Avoiding À-Trous Wavelet Transform (EAWT)[1] as an efficient and stable denoiser. This method uses a wavelet decomposition, where the input $c_0(p)$ is iteratively smoothed using a $B_3$-spline kernel $h$ [2]. At each level $i$, the signal is decomposed into a residual $c_{i+1}$ as follows:
>
> $$
> c_{i+1}(p) = c_i(p) * h_i,
> $$
>
> where $h_i$ expands its support by inserting $2^{i-1}$ zeros between coefficients at each step, ensuring computational efficiency.
>
> To preserve edges during smoothing, a data-dependent weighting function $w(p, q)$ is introduced. This function incorporates information from the ray-traced input image (rt), normal buffer ($n$), and position buffer ($x$):
>
> $$
> w(p, q) = w_{\text{rt}} \cdot w_n \cdot w_x,
> $$
>
> where
>
> $$
> w_{\text{rt}}(p, q) = \exp \left( -\frac{\| I_p - I_q \|^2}{\sigma_{\text{rt}}^2} \right)
> $$
>
> is based on color differences between pixels $p$ and $q$. The $\sigma$-parameters control the sensitivity to variations. These weights ensure that the filter adapts to the scene structure, preventing excessive edge blurring.
>
> The computation of $c_{i+1}(p)$ involves a normalization factor $k$, defined as:
>
> $$
> k = \sum_{q \in \Omega} h_i(q) \cdot w(p, q).
> $$
>
> Using $k$, $c_{i+1}(p)$ is then computed as:
>
> $$
> c_{i+1}(p) = \frac{1}{k} \sum_{q \in \Omega} h_i(q) \cdot w(p, q) \cdot c_i(q).
> $$
>
> We perform three iterations of this process, balancing accuracy and computational efficiency.
>
> [1] Dammertz, Holger, Sewtz, Daniel, Hanika, Johannes, and Lensch, Hendrik PA.
> "Edge-avoiding a-trous wavelet transform for fast global illumination filtering."
> *Proceedings of the Conference on High Performance Graphics*, 2010, pp. 67–75.
>
> [2] MURTAGH F.: Multiscale transform methods in data analysis.

---

> ### Comment · Reviewer_Cqns · 2024-11-20
>
> Dear authors,
>
> Thanks for your reply! I tentatively have no further questions. By the way, if I understand the policy correctly,  ICLR allows authors to upload a revised PDF during the discussion period. So maybe some visuals and detailed explanations could be updated to the PDF directly.

---

> > ### Author Response · Authors · 2024-11-25
> >
> > Dear reviewer,
> >
> > Thank you for your reply and valuable feedback! We understand your concerns and appreciate your guidance. At this stage, we are a bit cautious about modifying the PDF draft, as it might make the document appear somewhat disorganized. However, we promise the discussions and suggestions here will be incorporated into the final version to ensure clarity and completeness.
> > Thank you again for your time and support!

---

### Author Response · Authors · 2024-11-20

We extend our sincere thanks to the reviewers for their detailed and insightful feedback on our manuscript. We are grateful for their positive remarks on our results, which recognize the superior performance of our method compared to established baseline methods.

### Descriptions of Reservoir Sampling

We apologize for any confusion caused by the unclear description of reservoir sampling in Sec. 4.1, which may have been misleading. To clarify, we are including a more detailed explanation of reservoir sampling.

Our description consists of 2 parts, we firstly describe RIS, and then describe spatial-temporal reuse.

---
**Resampled Importance Sampling (RIS)**

According to the multiple importance sampling (MIS) theory [Veach & Guibas (1995)], the variance of the Monte Carlo estimator decreases when $p_{\text{dir}}(\omega_i)$ closely approximates the integrand $f(\omega_i)$. When $p_{\text{dir}}(\omega_i)$ is proportional to $L_{\text{env}}(\omega_i) f_r(x, \omega_i, d, m)$, the Monte Carlo estimator becomes highly efficient. However, directly sampling from such a distribution is infeasible due to its lack of a closed-form expression.

To address this, Resampled Importance Sampling (RIS) [1] provides a more advanced technique to approximate the distribution $p_\mathrm{dir}(\omega_i) \propto L \cdot f$, which is referred to as the **target distribution**. In RIS, we first choose an easy-to-sample distribution $q_{\text{dir}}(\omega)$ (referred to as the **proposal distribution**) and generate $m$ candidate samples $\mathcal{S} = \{\omega_1, ..., \omega_m\}$ from it. In our implementation, we use $L_{\text{env}}(\omega)$ as the proposal distribution. Then, we evaluate
$\hat{p}_{\text{dir}}(\omega_i) = L(\omega_i)\cdot f(\omega_i)$

for each candidate sample, and assign a weight

$$
\gamma_i = \frac{\hat{p_{\text{dir}}}(\omega_i)}{q_{\text{dir}}(\omega_i)}
$$

to each
$\omega_i \in \mathcal{S}$. Here, the hat in $\hat{p}$ means that it is *not necessarily a normalized PDF* (recall the fact that the normalized $p\propto\hat{p}$ is unknown).
Finally, we resample from the $\mathcal{S}$ according to the weight $\gamma_i$. Weighted-averaging the sampled results after repeating for $N$ times forms an $N$-sample RIS estimator of (which is Eq. 9 in our paper):


$$
C_{\text{PBR}}^{\text{dir}}(\mathbf{r}) \approx \frac{1}{N} \sum_{i=1}^N \left( \frac{f(\omega_i)}{\hat{p_{\text{dir}}}(\omega_i)} \frac{1}{m} \sum_{s=1}^m \frac{\hat{p_{\text{dir}}}(\omega_s)}{q_{\text{dir}}(\omega_s)} \right)
$$


Intuitively, the estimator behaves as if sampling directly from $p_{\text{dir}}$, weighted by the RIS sample weight to adjust for the difference between $p_{\text{dir}}$ and $q_{\text{dir}}$.

[1] Justin F Talbot. 2005. Importance resampling for global illumination. Brigham Young University.

---

**Spatial-temporal Reuse**

This equation can be implemented in a straightforward way by generating and storing all $m$ candidate samples before selecting a final sample. However, this approach is computationally demanding. To address this, we  utilize weighted reservoir sampling (WRS) [Chao 1982], which transforms RIS into a streaming way. That is, we maintain a reservoir structure $r$ for each pixel, with $r.y$ serving as the selected sample, $r.\gamma_{\text{sum}}$ serving as the sum of the weights, and $r.M$ representing the number of samples seen so far. When a new candidate $(w_s, \gamma_s)$ comes in, we update $r.\gamma_{\text{sum}}$ and $r.M$, then we decide whether to select it based on the ratio:

$$
\frac{\gamma_s}{r.\gamma_{\text{sum}}}.
$$

After the final sample is selected for each primary ray (or pixel), we exploit both spatial reuse and temporal reuse. Specifically, we merge reservoirs from neighboring pixels (spatial reuse) and previous frames (temporal reuse).

As $i$ increases, the influence of temporal reuse gradually extends to encompass contributions from all past $i$ frames. Similarly, spatial reuse progressively incorporates information from a larger region of the screen. This expansion occurs because, in each iteration, a pixel’s reservoir merges with information from its neighboring pixels, and in subsequent frames, those neighboring pixels' reservoirs have already integrated data from their own surroundings. Consequently, both spatial and temporal reuse effectively propagate information across increasingly broader spatial and temporal domains.

---

### Author Response · Authors · 2024-11-26
**Revised PDF**

Dear reviewers,

We sincerely thank you for your detailed and insightful feedback on our manuscript. As suggested by Reviewer Cqns, we have uploaded a revised version of our manuscript. To summarize, our revisions include:
1. **Updated description of reservoir sampling in Sec 4.1**: We acknowledge that the previous description of reservoir sampling is not clear enough, so we updated the description, mostly based on our top-level comment. We believe that it is more detailed and readable compared to the previous version.
2. **Description of denoiser in Appendix Sec A**: We apologize for not providing a detailed explanation of our adopted denoiser before, so we have added a paragraph in the appendix for it.
3. **Description of spatial-temporal reuse in Appendix Sec A**: Reviewer QNEc requires a more detailed description of our spatial-temporal reuse, so we have added a paragraph in the appendix for it.

Thank you again for your time and support!

---

### Meta-Review · Area_Chair_t2NH · 2024-12-20

**Metareview:**

This paper introduces an inverse rendering framework for the joint reconstruction of shape, material, and lighting. In the initial stage, it extracts a rough shape, followed by the second stage, where multi-bounce path tracing is employed to estimate indirect illumination more effectively. The use of path tracing combined with reservoir sampling results in notable improvements in reconstruction quality.

Reviews raised some issues, and the rebuttal has effectively addressed most issues. The remaining major issues primarily revolve around the contributions of the proposed method, the need for comparison with Neural-PBIR, and additional experiments on more datasets.

Regarding the experiments, more comprehensive evaluations on the Stanford-ORB dataset and a thorough comparison with Neural-PBIR would strengthen the paper. During the rebuttal period, the authors provided comparisons with Neural-PBIR on a limited number of scenes. The results suggest that the proposed method demonstrates advantages over Neural-PBIR. While the comparison is not exhaustive, given the limited rebuttal timeframe and the unavailability of Neural-PBIR's code, the results are convincing enough to support the claims.

As for the contributions, directly adopting ReSTIR does not constitute a major contribution. However, the authors emphasize that their primary contribution lies in proposing a holistic inverse rendering framework that cohesively integrates all components. The experiments seem to support the effectiveness of this framework compared to prior methods.

**Additional Comments On Reviewer Discussion:**

The reviews highlighted several issues, including the dependency on high-quality initialization, the focus on primary rays only, long optimization times, the lack of results demonstrating specular reflection, and the choice to use triangle meshes. The rebuttal effectively addressed most of these concerns. The remaining major issues primarily relate to clarifying the contributions of the proposed method, providing comparisons with Neural-PBIR, and conducting additional experiments on more datasets. While these issues were not resolved perfectly, the rebuttal addressed them sufficiently.

---

> ### Public Comment · ~Yuxin_Dai1 · 2025-02-06
>
> Dear Area Chairs and Program Chairs,
>
> Thank you for overseeing the review process of our paper.
>
> Our current title, "Inverse Rendering for Shape, Light, and Material Decomposition using Multi-Bounce Path Tracing and Reservoir Sampling," is quite lengthy. Since inverse rendering inherently refers to shape, light, and material decomposition, we propose a more concise title:
> "Inverse Rendering with Multi-Bounce Path Tracing and Reservoir Sampling."
>
> We kindly seek your approval for this change. Thank you for your time and consideration.
>
> Best regards,
>
> The authors

---

> > ### Comment · Area_Chair_t2NH · 2025-02-07
> >
> > Dear authors,
> >
> > After a discussion with SAC and PCs, your request for title change has been approved by PCs.
> >
> > AC

---

### Decision · Program_Chairs · 2025-01-22

Accept (Poster)